# Strategy-driven Bidirectional Efficient Adaptive Federated Learning

**Abstract**

We propose two new strategies within the framework of federated learning: NLA (New Lazy Aggregation) and AA (Accelerated Aggregation). The NLA strategy reduces the costs of communication and computation by adaptively skipping the gradient, and the AA accelerates the computation and reduces the communication cost by adaptively accruing the gradient computation. We propose six strategy-driven communication-efficient non-convex adaptive federated learning algorithms via NLA and AA. In particular, based on these novel strategies and compression techniques, we propose two new algorithms: FedBNLACA and FedBACA, to reduce bidirectional communication costs. We give a theoretical guarantee of these algorithms for client (full or partial) participation in correlation under a non-convex setting. In non-convex stochastic optimization full client participation setting, our proposed FedBNLACA and FedBACA algorithms achieve the same convergence rate as its non-compressed counterpart. We have demonstrated through extensive experiments that our protocol achieves efficient training in non-convex environments and is robust to a large amounts of devices, partial participation, and unbalanced data.

**Keywords:** Federated learning, non-convex optimization, heterogeneous data, bidirectional communication, adaptive gradient accumulation, robustness.

**Mathematics Subject Classification (2020):** 62XXX

## 1 Introduction

Federated Learning (FL) is an emerging approach in machine learning (ML) that enables collaborative model training while maintaining data privacy by avoiding the need to share raw data. This approach ensures that data privacy is protected, as data does not need to leave the owner's possession (McMahan et al., 2017). Federated learning (FL) models entail clients or devices, including smartphones and personal computers (PCs), independently training local machine learning models utilizing their own data without exchanging it. These clients do not send raw data to central servers (such as cloud servers), but update model parameters locally (such as smartphone, PC updates). The server then generates a shared global ML model by aggregating local parameter updates. In Federated learning (FL), conventional distributed stochastic gradient descent (SGD) proves inadequate for complex scenarios characterized by non-independent and non-identically distributed data across clients, as well as limited client participation in each communication round. To address these challenges, various federated optimization methods utilize client-side updates. In particular, clients execute several local model update iterations

before transmitting information to the central server, which significantly diminishes the communication burden associated with model training. A notable instance of this strategy is the FedAvg algorithm proposed by (McMahan et al., 2017), where the global model is refined by aggregating multiple local stochastic gradient descent (SGD) update steps performed by the participating clients.

Despite its ability to enable training without direct data sharing and yield competitive performance, the FedAvg algorithm still faces notable challenges in practical federated learning scenarios. 1) Lack of adaptivity. The underlying stochastic gradient descent (SGD) updates may be ill-suited for handling gradient noise that follows a heavy-tailed distribution common characteristic when training large-scale models (Devlin et al., 2019; Brown et al., 2020). 2) Unaffordable communication. Frequent synchronization between clients and the server results in substantial uplink and downlink transmission costs, some of which involve redundant or non-essential parameters.

Many studies have addressed the aforementioned issues. 1) To enhance adaptivity in the federated learning framework, the FedAdam algorithm and its variants (Tong et al., 2020; Wang et al., 2022b; Wu et al., 2023) have been proposed, integrating the adaptive gradient method. 2) In recent efforts to lower communication costs, three strategies have emerged: (i) Local optimization (Huang et al., 2023; Li and Wang, 2019; Li et al., 2019b; Mishchenko et al., 2022; Karimireddy et al., 2019a), where clients perform multiple optimization steps before transmitting data to the server, thus decreasing the need for extensive global communication and overall reducing communication costs. (ii) Compression techniques (Reisizadeh et al., 2020; Chen et al., 2021; Richtárik et al., 2023; Beznosikov et al., 2023), where clients transmit compressed data during each communication round, minimizing the communication load through various compression strategies. (iii) Lazy aggregation (Chen et al., 2018; Sun et al., 2019; Ghadikolaei et al., 2021; Mishchenko et al., 2022), in which certain parameters may be adjusted. Some parameters may be similar to those of the previous round, so they do not need to be transmitted (i.e., the parameters of the previous round are used in the current round), which also effectively reduces the communication cost.

However, although the lazy aggregation algorithm provides a strategy to reduce communication costs, it is more difficult to implement in practice, especially in joint learning where only some clients are involved in the training. While previous studies (Sun et al., 2020; Wang et al., 2022a; Gruntkowska et al., 2022; Fatkhullin et al., 2021) have explored bidirectional algorithms, they have not addressed the issue of heterogeneity in federated learning (FL), the participation of certain servers in the training process, nor the associated theoretical guarantees. Additionally, their proposed methods for reducing communication costs are challenging to implement in scenarios involving heterogeneous partial client participation. A natural thought is:

**How do we design a simple and more efficient strategy to improve communication efficiency based on bidirectional communication?**

In this paper, we first present two novel updated strategies (NLA and AA), apply them to the FedAMS and FedCAMS algorithms (Wang et al., 2022b), and then develop bidirectional communication-efficient adaptive algorithms under a non-convex setting.

**Main contributions**

• We propose two new communication strategies for federated learning, referred to as NLA and AA, and incorporate them into adaptive federated optimization to obtain FedNLAA and FedAA. Inspired by but distinct from LAG methods (Chen et al., 2018; Sun et al., 2019; Mishchenko et al., 2022), NLA and AA are formulated as new strategies acting on the actually transmitted object, such as a local model update, a compressed update, or a downlink parameter. NLA introduces a last-iterate relative-stability rule, while AA introduces a conditional accumulation rule. These strategy-level designs are simple to implement and are naturally compatible with adaptive optimization, compression, bidirectional communication, and partial client participation.

• We design two strategy-driven efficient non-convex adaptive federated learning algorithms: FedNLACA and FedACA. Both algorithms enhance communication efficiency and adaptability in federated learning. They significantly reduces communication costs while maintaining high accuracy by employing error feedback, compression strategies, and our proposed strategy, respectively. It is important to emphasize that both FedNLACA and FedACA achieve a convergence rate of $\mathcal{O}(1/T)$.

• We have further developed two strategy-driven bidirectional efficient non-convex adaptive federated learning algorithms: FedBNLACA and FedBACA, which employ two strategic approaches to facilitate a reduction in bidirectional communication costs for both uplink and downlink. A convergence analysis was performed under conditions of general non-convexity and data heterogeneity, demonstrating that both FedBNLACA and FedBACA can also attain a convergence rate of $\mathcal{O}(1/T)$.

• Extensive experiments on various benchmarks showed that our proposed algorithms are well adaptive in training real-world machine learning models.

## 2 Related work

**Adaptive Gradient Methods** Adaptive gradients (Zeiler, 2012; Duchi et al., 2011; Kingma and Ba, 2014; Reddi et al., 2018) are a series of algorithms that effectively reduce the relatively slow convergence and over-sensitivity to parameters of gradient descent methods in the face of heavy-tailed stochastic gradients, and are heavily used to train large networks.

**Federated Learning** FedAvg was introduced by (McMahan et al., 2017) as the inaugural algorithm for federated learning (FL). By employing periodic model averaging, this approach significantly mitigates communication overheads. Initial studies focused on the analysis of FL algorithms within a homogeneous data framework, while more recent investigations have expanded the scope of federated learning to encompass heterogeneous data environments (non-iid) and non-convex models (Li and Wang, 2019; Li et al., 2019a; Sahu et al., 2018; Yang et al., 2019; Wang et al., 2022b). (Li et al., 2019a; Sahu et al., 2018) introduced the FedProx and FedDANE algorithms to address heterogeneity in federated optimization. (Wang et al., 2022b) gave theoretical results of data heterogeneous federated learning under the conditions of adaptive methods. In this paper, we follow the ideas of Wang et al. (2022b). Numerous studies have been conducted that build upon the FedAvg framework, including notable contributions such as FedNova (Wang et al., 2020), and SCAFFOLD (Karimireddy et al., 2019a), as well as various other adaptations of FedAvg (Yang et al., 2021; Wang et al., 2022b). More recently, (Reddi et al., 2020) proposed

several adaptive federated optimization algorithms namely FedAdagrad, FedYogi, and FedAdam designed to mitigate convergence issues inherent in the FedAvg approach. Additionally, (Chen et al., 2020) presented Local AMSGrad, while (Tong et al., 2020) introduced a suite of federated adaptive gradient methods that incorporate calibration mechanisms.

**Effective methods of communication** Many methods to reduce communication costs have been proposed in federated learning, three main ideas are 1) multiple rounds of local iteration methods (Elgabli et al., 2022; McMahan et al., 2017; Li et al., 2019a; Sahu et al., 2018), 2) compression and error feedback (Reisizadeh et al., 2020; Elgabli et al., 2022; Haddadpour et al., 2020; Wang et al., 2022b), 3) lazy aggregation (Sun et al., 2020; Mao et al., 2022; Chen et al., 2018; Sun et al., 2019).

The first line of work reduces communication frequency through local updates or adaptive communication scheduling. In addition to FedAvg local update methods, AdaComm-type methods adapt the communication period according to the training stage or optimization progress (Wang and Joshi, 2019). Related dynamic-threshold or $\Delta$-thresholding communication rules trigger communication only when the change of the local update or model difference is sufficiently large. These methods are effective in reducing the number of synchronization rounds, but their main control variable is usually the communication period or synchronization frequency.

The second line of work reduces the size of each transmitted message through quantization, compression, or sparsification. Representative examples include Top-$k$ sparsification, sparsified SGD with memory, Deep Gradient Compression, and adaptive sparsification methods (Stich et al., 2018; Lin et al., 2017). These methods mainly determine which gradient coordinates, residuals, or update components should be transmitted. Error-feedback methods further improve compressed communication by accumulating compression residuals and injecting them into subsequent communication rounds (Richtárik et al., 2023). These approaches are powerful for message-size reduction, but their communication decision is mainly coordinate-level or residual-level rather than strategy-level over the whole transmitted update.

The third line of work is lazy, stale-gradient, or event-triggered communication. LAG and LAQ reduce communication by reusing stale gradient information or stale quantized gradient information when the local variation is sufficiently small (Chen et al., 2018; Sun et al., 2019). Self-triggered upload methods, such as LENA, allow clients to decide whether fresh gradient information is necessary (Ghadikolaei et al., 2021). Stale-gradient handling in asynchronous SGD is also related, since asynchronous methods need to decide how to use, compensate, delay, or discard outdated gradient information caused by staleness (Zheng et al., 2017; Dutta et al., 2018). These methods provide important event-triggered communication principles, but they mainly focus on stale-gradient reuse, stale-gradient discarding, or upload-side triggering.

Compared with the above communication-efficient methods, the proposed NLA and AA strategies operate at a different level. Adaptive communication methods such as AdaComm mainly adjust the communication period; DGC and sparsification methods mainly decide which coordinates or residuals should be transmitted; and LAG/LAQ mainly control stale gradient or stale quantized gradient information. In contrast, NLA and AA act directly on the actually transmitted object, such as a local update, a compressed update, or a downlink parameter. NLA uses a last-iterate relative-stability rule to suppress redundant transmissions, while

AA introduces a conditional accumulation rule to exploit stable consecutive updates at the communication-strategy level.

However, most of the above methods are only unidirectional (uplink), and although Sun et al. (2020); Wang et al. (2022a); Gruntkowska et al. (2022); Fatkhullin et al. (2021) considered bidirectional algorithms, they did not consider the heterogeneity problem in FL, the involvement of some of the servers in the training as well as the related theoretical guarantee, and the adaptive problem. Moreover, their strategies to reduce the communication cost are difficult to implement under heterogeneous partial clients participation. Therefore, how to give simple strategies to reduce communication cost, and how to design adaptively efficient algorithms for bidirectional communication with partial clients participation is the focus of this paper.

**Notation** Let $\mathbf{x}, \mathbf{y} \in \mathbb{R}^d$ be vectors. The notations $\sqrt{\mathbf{x}}$, $\mathbf{x}^2$, and $\mathbf{x}/\mathbf{y}$ represent the element-wise square root, square, and division operations applied to the vectors, respectively. For a vector $\mathbf{x}$ and a matrix $A$, the symbol $\|\cdot\|$ denotes the $\ell_2$-norm, such that $\|\mathbf{x}\| = \|\mathbf{x}\|_2$ and $\|A\| = \|A\|_2$. In the context of algorithms, the variable $t$ indicates the iteration index, while $m$ denotes the total number of clients or devices involved.

## 3 Preliminary

This paper aims to study the federated learning non-convex optimization problem, which is formulated as follows:

$$\min_{\theta \in \mathbb{R}^d} f(\theta) = \frac{1}{m} \sum_{i=1}^{m} F_i(\theta),$$

where $F_i(\theta) = \mathbb{E}_{\xi \sim \mathcal{D}_i} F_i(\theta, \xi_i), F_i(\theta)$ denotes the local non-convex loss, $\mathcal{D}_i$ represents the data distribution on $i$-th clients. $m$ represents the number of all clients, $d$ denotes the dimension of the model parameters. In the non i.i.d setting, distributions $\mathcal{D}_i$ and $\mathcal{D}_j$ can vary from each other, i.e., $\mathcal{D}_i \neq \mathcal{D}_j$, $\forall i \neq j$. In the stochastic setting, one can only get unbiased estimates of $F_i(\theta)$, i.e., the stochastic gradient $\mathbf{g}_t^i = \nabla F_i(\theta, \xi_i)$.

### 3.1 Assumption and Compressor

In this subsection, we introduce several assumption for establishing convergence analysis of our proposed algorithms, and biased compressor for designing algorithms.

**Assumption 1.** *(Smoothness).There exists a constant $L_i$ such that the loss function associated with the $i$-th worker, denoted by $F_i(\boldsymbol{\theta})$, satisfies the following inequality for all $\mathbf{x}, \mathbf{y} \in \mathbb{R}^d$:*

$$|F_i(\mathbf{x}) - F_i(\mathbf{y}) - \langle \nabla F_i(\mathbf{y}), \mathbf{x} - \mathbf{y} \rangle| \leq \frac{L_i}{2} \|\mathbf{x} - \mathbf{y}\|^2.$$

This condition implies that $F_i$ is smooth and equivalently satisfies the $L_i$-gradient Lipschitz continuity condition, expressed as $\|\nabla F_i(\mathbf{x}) - \nabla F_i(\mathbf{y})\| \leq L_i \|\mathbf{x} - \mathbf{y}\|$. Such an assumption is commonly adopted in the literature (see, e.g., Yang et al. (2021); Reddi et al. (2018)).

**Assumption 2.** *(Bounded Gradient). For each worker $i$, the loss function $F_i(\theta)$ possesses a stochastic gradient that is bounded in the $\ell_2$-norm by a constant $G$. Formally, for all random*

variables $\xi$, it holds that $\mathbb{E}\|\nabla f_i(\theta, \xi)\| \leq G$. Furthermore, it is assumed that the parameter vector satisfies $\|\theta\| \leq H$. This bounded gradient condition is a common assumption in the analysis of adaptive gradient methods Reddi et al. (2018).

**Assumption 3.** *(Bounded Variance). The bounded local variance, i.e. for all $\theta$, $i \in [m]$, $\mathbb{E}[\|\nabla f_i(\theta, \xi) - \nabla F_i(\theta)\|^2] \leq \sigma_l^2$; and global variance constraint, i.e. $\frac{1}{m}\sum_{i=1}^{m}\|\nabla F_i(\theta) - \nabla f(\theta)\|^2 \leq \sigma_g^2$, where $\sigma_l^2$ and $\sigma_g^2$ are some positive constants.*

The bounded variance assumption is widely adopted in adaptive gradient algorithms Yang et al. (2021); Reddi et al. (2020). Specifically, the bounded local variance characterizes the inherent stochasticity in gradient estimates, while the bounded global variance reflects the degree of data heterogeneity across clients. It is important to note that these variances are assumed to be finite. In particular, a global variance of $\sigma_g = 0$ corresponds to the i.i.d. setting, where the datasets across all clients follow the same distribution.

**Assumption 4.** *(Compression–Aggregation Compatibility). Consider a biased compressor . There exists a constant $\gamma > 0$ such that for every iteration $t \geq 0$, the following inequality is satisfied:*

$$\left\| \mathcal{C}\left(\frac{1}{m}\sum_{i=1}^{m}[\Delta_t^i + \mathbf{e}_t^i]\right) - \frac{1}{m}\sum_{i=1}^{m}\mathcal{C}(\Delta_t^i + \mathbf{e}_t^i) \right\| \leq \gamma \left\|\frac{1}{m}\sum_{i=1}^{m}\Delta_t^i\right\| \tag{3.1}$$

*and*

$$\left\|\frac{1}{M_t}\sum_{i \in M_t}\mathcal{C}(q_t^i)\right\| \leq \lambda \left\|\frac{1}{m}\sum_{i=1}^{m}\Delta_t^i\right\|. \tag{3.2}$$

Here $M_t$ denotes the set of all clients satisfying (3.5) or (3.6) at round $t$, and $\mathcal{C}(q_t^i)$ is given in Algorithms 2 and 3. This compatibility condition is used only to control the mismatch between compressing the averaged update and averaging the compressed client updates. It is not implied by the standard biased-compressor condition.

**Remark 1.** *Assumption 4 is a compression–aggregation compatibility condition. It is stronger than the standard biased-compressor contraction condition and should not be understood as a generic property of all biased compressors. In particular, although Top-k satisfies $\|C(x) - x\| \leq \sqrt{1 - k/d}\|x\|$, this contraction property alone does not imply Assumption 4, because Top-k is nonlinear and the selected supports may differ before and after aggregation. Thus, the convergence theory for the compressed algorithms applies to compressors and optimization trajectories satisfying this compatibility condition. For the Top-k compressor used in our experiments, we empirically verify the associated trajectory-level mismatch ratios in Appendix 1.1.*

**Definition 1.** *(Biased Compressor). A possibly biased operator $\mathcal{C} : \mathbb{R}^d \rightarrow \mathbb{R}^d$ is considered. There exists a constant $q \in [0, 1]$ such that for any vector $\theta \in \mathbb{R}^d$, the following inequality holds:*

$$\|\mathcal{C}(\theta) - \theta\| \leq q\|\theta\|.$$

A value of $q = 0$ implies that $\mathcal{C}$ is an identity mapping, i.e., no compression is applied to $\theta$. Notable examples of such compressors include the scaled-sign compressor and the top-$k$ compressor.

**Top-$k$ Compressor** (Shi et al., 2019; Stich et al., 2018). For a given integer $k$ with $1 \leq k \leq d$, consider any vector $\theta \in \mathbb{R}^d$. Its coordinates are ordered by their magnitude, denoted as $|\theta_{(1)}| \leq |\theta_{(2)}| \leq \cdots \leq |\theta_{(d)}|$. Let $h_1, h_2, \ldots, h_d$ be the standard unit basis vectors of $\mathbb{R}^d$. The top-$k$ compressor $\mathcal{C}_{\text{top}} : \mathbb{R}^d \to \mathbb{R}^d$ is then defined as:

$$\mathcal{C}_{\text{top}}(\theta) = \sum_{i=d-k+1}^{d} \theta_{(i)} h_{(i)},$$

which retains only the $k$ coordinates of $\theta$ with the largest magnitudes. Define the compression ratio as $r = k/d$. It can be shown that $\|C_{\text{top}}(\theta) - \theta\|^2 \leq (1 - k/d)\|\theta\|^2 = (1 - r)\|\theta\|^2$, and thus we have $q = \sqrt{1 - r}$.

**Scaled-sign Compressor** (Karimireddy et al., 2019b). For $1 \leq k \leq d$ and $\forall \theta \in \mathbb{R}^d$, the compressor $\mathcal{C}_{\text{sign}} : \mathbb{R}^d \to \mathbb{R}^d$ is defined as

$$\mathcal{C}_{\text{sign}}(\theta) = \|\theta\|_1 \cdot \text{sign}(\theta)/d.$$

For scaled sign compressor, when $\|C_{\text{sign}}(\theta) - \theta\|^2 = (1 - \|\theta\|_1^2/d\|\theta\|^2)\|\theta\|^2$, thus $q = \sqrt{1 - \|\theta\|_1^2/d\|\theta\|^2}$.

## 3.2 Two Strategies: NLA and AA

Prior to presenting the new strategies, it is essential to examine the LAG algorithm (Chen et al., 2018; Sun et al., 2019; Mishchenko et al., 2022):

$$\left\|\nabla F_m(\boldsymbol{\theta}_m^{t-1}) - \nabla F_m(\boldsymbol{\theta}^t)\right\|^2 \leq \frac{1}{\alpha^2 m^2} \sum_{r=1}^{R} \xi_r \left\|\boldsymbol{\theta}^{t+1-r} - \boldsymbol{\theta}^{t-r}\right\|^2, \tag{3.3}$$

$$L_m^2 \left\|\boldsymbol{\theta}_m^{t-1} - \boldsymbol{\theta}^t\right\|^2 \leq \frac{1}{\alpha^2 m^2} \sum_{r=1}^{R} \xi_r \left\|\boldsymbol{\theta}^{t+1-r} - \boldsymbol{\theta}^{t-r}\right\|^2. \tag{3.4}$$

While the aforementioned concept is innovative, it necessitates the establishment of parameters for the initial R iterations. Determining the appropriate value for R poses a challenge, and furthermore, the inclusion of parameters from all R iterations in the computation may lead to complications in data storage. Therefore, we propose two new aggregation strategies. The $S_t$ denotes the sum of all participating training clients at the $t$-th iteration, $C, D, \alpha$ are some postive constants, $m$ represents the number of all clients.

**(1) NLA Strategy** (**N**ew **L**azy **A**ggregation). For any **x**, let $\rho_t = \mathbf{x}_t - \mathbf{x}_{t-1}$. If

$$\|\rho_t\| \leq \frac{C}{\alpha S_t}\|\mathbf{x}_{t-1}\| : \mathbf{x}_t \leftarrow \mathbf{x}_{t-1}, else : \mathbf{x}_t \leftarrow \mathbf{x}_t. \tag{3.5}$$

**Example 1:** Let $q_t^i = \Delta_t^i - \Delta_{t-1}^i$, if $\|q_t^i\| \leq \frac{C}{\alpha S_t}\|\Delta_{t-1}^i\|, i \in M_t : \tilde{\Delta}_t^i \leftarrow \Delta_{t-1}^i$, else: $\tilde{\Delta}_t^i \leftarrow \Delta_t^i$, the specific parameters are given in Algorithm 1.

**Example 2:** Let $\mathcal{C}(q_t^i) = \widehat{\Delta}_t^i - \widehat{\Delta}_{t-1}^i$, if $\|\mathcal{C}(q_t^i)\| \leq \frac{C}{\alpha S_t}\|\widehat{\Delta}_{t-1}^i\|, i \in M_t : \widehat{\tilde{\Delta}}_t^i \leftarrow \widehat{\Delta}_{t-1}^i$, else:

$\widehat{\widetilde{\Delta}}_t^i \leftarrow \widehat{\Delta}_t^i$, the specific parameters are given in Algorithm 2.

**Example 3:** Let $\mathcal{C}(Q_t^i) = \widehat{\theta}_t^i - \widehat{\theta}_{t-1}^i$, if $\|\mathcal{C}(Q_t^i)\| \le \frac{C}{\alpha S_t}\|\widehat{\theta}_{t-1}^i\|, i \in M_t : \widehat{\widetilde{\theta}}_t^i \leftarrow \widehat{\theta}_{t-1}^i$, else: $\widehat{\widetilde{\theta}}_t^i \leftarrow \widehat{\theta}_t^i$, the specific parameters are given in Algorithm 3.

**Remark 2.** *The NLA Strategy presented herein represents a modification of the lazy aggregation method as described in previous works (Chen et al., 2018; Sun et al., 2019). This approach is characterized by its operational simplicity, necessitating only a comparison with the parameters from the preceding iteration. The NLA Strategy, as a novel lazy aggregation method, effectively reduces communication overhead by decreasing the quantity of parameters that need to be transmitted. (The NLA strategy assesses whether the difference between the parameters from the $(t-1)$th and $t$-th iterations falls within a minimal threshold, indicating that these parameters are closely aligned. If this proximity is confirmed, the parameter from the $(t-1)$th iteration is retained in place of the parameter from the $t$-th iteration.)*

**Remark 3** (Relation between NLA and LAG). *NLA is related to LAG-type lazy aggregation methods in the sense that both use a communication-triggering rule to decide whether stale information can be reused. However, NLA differs from LAG in the quantity controlled by the triggering condition. LAG-type rules are mainly designed to control stale-gradient discrepancies, either directly or through smoothness-based surrogate bounds. By contrast, NLA controls the relative variation of the actually transmitted object, which can be a local update, a compressed update, or a downlink model parameter. Therefore, even when LAG is instantiated with a single-step window ($R = 1$), NLA is not simply a one-step LAG rule. A detailed comparison between NLA and LAG, including the case $R = 1$, is provided in Appendix 1.3.*

**(2) AA Strategy** (**A**ccelerated **A**ggregation). For any $\mathbf{x}$, let $\rho_t = \mathbf{x}_t - \mathbf{x}_{t-1}$. if

$$\|\rho_t\| \le \frac{D}{\alpha S_t}\|\mathbf{x}_{t-1}\| : \mathbf{x}_t \leftarrow \mathbf{x}_{t-1} + \mathbf{x}_t, else : \mathbf{x}_t \leftarrow \mathbf{x}_t. \tag{3.6}$$

**Example 4:** Let $q_t^i = \Delta_t^i - \Delta_{t-1}^i$, if $\|q_t^i\| \le \frac{D}{\alpha S_t}\|\Delta_{t-1}^i\|, i \in M_t : \tilde{\Delta}_t^i \leftarrow \Delta_{t-1}^i + \Delta_t^i$, else: $\tilde{\Delta}_t^i \leftarrow \Delta_t^i$, the specific parameters are given in Algorithm 1

**Example 5:** Let $\mathcal{C}(q_t^i) = \widehat{\Delta}_t^i - \widehat{\Delta}_{t-1}^i$, if $\|\mathcal{C}(q_t^i)\| \le \frac{D}{\alpha S_t}\|\widehat{\Delta}_{t-1}^i\|, i \in M_t : \widehat{\widetilde{\Delta}}_t^i \leftarrow \widehat{\Delta}_{t-1}^i + \widehat{\Delta}_t^i$, else: $\widehat{\widetilde{\Delta}}_t^i \leftarrow \widehat{\Delta}_t^i$, the specific parameters are given in Algorithm 2.

**Example 6:** Let $\mathcal{C}(Q_t^i) = \widehat{\theta}_t^i - \widehat{\theta}_{t-1}^i$, if $\|\mathcal{C}(Q_t^i)\| \le \frac{D}{\alpha S_t}\|\widehat{\theta}_{t-1}^i\|, i \in M_t : \widehat{\widetilde{\theta}}_t^i \leftarrow \widehat{\theta}_{t-1}^i + \widehat{\theta}_t^i$, else: $\widehat{\widetilde{\theta}}_t^i \leftarrow \widehat{\theta}_t^i$, the specific parameters are given in Algorithm 3.

**Remark 4.** *The AA strategy is a novel acceleration method designed to reduce communication costs by speeding up the process. The proposal of this strategy is based on a fundamental motivation: when the parameters of the $(t-1)$th iteration are very close to the parameters of the $t$th iteration, it is possible to achieve an update of both steps at once (i.e., by adding the parameters of the $(t-1)$th and $t$th iterations). This core idea is consistent with the principles of the NLA algorithm. Through adaptive accelerated iterative descent, the AA strategy can effectively reduce communication costs. It is worth noting that even without the AA strategy, the iterative process*

*of conventional algorithms can still achieve results similar to those of the AA strategy, but a more in-depth analysis of the AA strategy will be reserved for future research.*

# 4 Strategy-driven Efficient Federated Learning

## 4.1 Federated New Lazy Aggregation AMSGrad and Federated Acceleration AMSGrad

In this section, we introduce two novel adaptive algorithm frameworks: **Fed**erated **N**ew **L**azy **A**ggregation **A**MSGrad (FedNLAA) and **Fed**erated **A**ccelerated **A**MSGrad (FedAA). In both algorithms, $\theta_t$ denotes the global model parameters at iteration $t$. At the beginning of iteration $t$, each participating client $i$ in the selected subset $S_t$ (of size $n$) receives the current global model $\theta_t$ from the server, i.e., $\theta_{t,0}^i = \theta_t$. The client then performs $K$ steps of local SGD updates with a local learning rate $\eta_l$, yielding the local model $\theta_{t,K}^i$. After computing the local model difference $\Delta_t^i = \theta_{t,K}^i - \theta_t$, the client determines whether it satisfies the respective strategy-NLA for FedNLAA or AA for FedAA-and subsequently transmits the resulting judged model difference $\widehat{\Delta}_t^i$ to the server. The server then aggregates these contributions by averaging to obtain the global model difference $\Delta_t$, which is used to update the global model. The complete procedures for FedNLAA and FedAA are detailed in Algorithm 1.

## 4.2 Convergence Analysis for FedNLAA and FedAA

**Full Participation:** All clients participate in training, i.e., $|\mathcal{S}_t| = m, \forall t \in [T]$.

**Theorem 1.** *Under Assumptions 1-3, if learning rate $\eta_l$ satisfies the following condition:*

$$\eta_l \leq \min\left\{\frac{1}{8KL}, \frac{\epsilon}{K_1[K_2 + K_3]}\right\},$$

*where $K_1 = K\sqrt{\beta_2 K^2 G^2 + \epsilon}$, $K_2 = (3 + C_1^2)\eta L$ and $K_3 = 2\sqrt{2(1 - \beta_2)}G$, then FedNLAA in Algorithm 1 under the full participation has*

$$\min_{t \in [T]} \mathbb{E}[\|\nabla f(\theta_t)\|^2] \leq K_4\left[\frac{f_0 - f_*}{\eta \eta_l K T} + \frac{\Xi}{T} + \Omega\right],$$

*where $K_4 = 4\sqrt{\beta_2 \eta_l^2 K^2 G^2 + \epsilon}$, $\Xi = \frac{C_1 G^2 d}{\sqrt{\epsilon}} + \frac{2C_1^2 \eta \eta_l K L G^2 d}{\epsilon}$, $\Omega = \frac{5\eta_l^2 K^2 L^2}{\sqrt{2\epsilon}}(\sigma_l^2 + 6K\sigma_g^2) + (3 + C_1^2)\eta^2 L + 2\sqrt{2(1 - \beta_2)}\eta G](\frac{2\eta_l}{m\eta\epsilon}\sigma_l^2 + \frac{2KC^2\eta_l G^2}{\alpha^2 \eta m^2 \epsilon}) + \frac{\sqrt{2}GC}{\alpha m\epsilon}$, and $C_1 = \frac{\beta_1}{1 - \beta_1}$.*

**Theorem 2.** *Under Assumptions 1-3, take learning rate $\eta_l$ as the one in Theorem 1, then FedAA in Algorithm 1 under the full participation has*

$$\min_{t \in [T]} \mathbb{E}[\|\nabla f(\theta_t)\|^2] \leq K_4\left[\frac{f_0 - f_*}{\eta \eta_l K T} + \frac{\Xi}{T} + \Omega\right],$$

*where $K_4$ is defined in Theorem 1, $\Xi = \frac{C_1 G^2 d}{\sqrt{\epsilon}} + \frac{2C_1^2 \eta \eta_l K L G^2 d}{\epsilon}$, $\Omega = \frac{5\eta_l^2 K^2 L^2}{\sqrt{2\epsilon}}(\sigma_l^2 + 6K\sigma_g^2) + (3 + C_1^2)\eta^2 L + 2\sqrt{2(1 - \beta_2)}\eta G](\frac{2\eta_l}{m\eta\epsilon}\sigma_l^2 + \frac{2K\eta_l G^2}{\eta\epsilon}) + \frac{\sqrt{2}G}{\epsilon}$, and $C_1 = \frac{\beta_1}{1 - \beta_1}$.*

**Algorithm 1:** FedNLAA and FedAA

**Input:** Initial value $\theta_1$, local step size $\eta_l$, global step size $\eta$, constants $\beta_1$, $\beta_2$ and $\epsilon$, $\Delta_0^i = 0$.

**Initialization:** $\mathbf{m}_0 \leftarrow 0, \mathbf{v}_0 \leftarrow 0$.

**for** $t = 1$ **to** $T$ **do**

    Server randomly selects a subset of clients $S_t$ and transmits $\theta_t$ to the subset of clients $S_t$.

    $\theta_{t,0}^i \leftarrow \theta_t$.

    **for** *each client* $i \in S_t$ *in parallel* **do**

        **for** $k = 0, ..., K-1$ **do**

            • Compute local stochastic gradient:

            $\mathbf{g}_{t,k}^i = \nabla F_i(\theta_{t,k}^i; \xi_{t,k}^i)$,

            • Update $\theta_{t,k+1}^i = \theta_{t,k}^i - \eta_l \mathbf{g}_{t,k}^i$.

        **end**

        • Compute $\Delta_t^i = \theta_{t,K}^i - \theta_t$ , $q_t^i = \Delta_t^i - \Delta_{t-1}^i$, Judges: If $q_t^i$ satisfies NLA (Example 1) or AA (Example 4).

        • Outputs: $\tilde{\Delta}_t^i$ .

    **end**

    Server aggregates: $\tilde{\Delta}_t = \frac{1}{|S_t|} \sum_{i \in \mathcal{S}_t} \tilde{\Delta}_t^i$, Update: $\mathbf{m}_t = \beta_1 \mathbf{m}_{t-1} + (1 - \beta_1)\tilde{\Delta}_t$, Update:

    $\mathbf{v}_t = \beta_2 \mathbf{v}_{t-1} + (1 - \beta_2)\tilde{\Delta}_t^2$,

    $[\hat{\mathbf{v}}_t = \max(\hat{\mathbf{v}}_{t-1}, \mathbf{v}_t, \epsilon)$ and $\theta_{t+1} = \theta_t + \eta \frac{\mathbf{m}}{\sqrt{\hat{\mathbf{v}}_t}}]$,

    or

    $[\hat{\mathbf{v}}_t = \max(\hat{\mathbf{v}}_{t-1}, \mathbf{v}_t)$ and $\theta_{t+1} = \theta_t + \eta \frac{\mathbf{m}}{\sqrt{\hat{\mathbf{v}}_t} + \epsilon}]$.

**end**

**Remark 5.** *Under the condition that $C = D$ and $\frac{C}{\alpha m} = 1$, the result in Theorem 1 reduces to that in Theorem 2. The upper bound for $\min_{t \in [T]} \mathbb{E}[\|\nabla f(\theta_t)\|^2]$ consists of three components. The first two terms decay as $T$ increases and vanish as $t$ goes to infinity. The last term depends on the local stochastic variance $\sigma_l$ and the global variance $\sigma_g$, which captures the degree of data heterogeneity. In the i.i.d. setting where all clients share the same data distribution, i.e., $\sigma_g = 0$, the variance term $\Omega$ becomes smaller.*

**Corollary 1.** *Suppose choose local learning rate $\eta_l = \Theta(\frac{1}{\sqrt{TK}})$ and global learning rate $\eta = \Theta(\sqrt{Km})$, when $T$ is sufficiently large, i.e., $T \geq Km$, the convergence rate for FedNLAA and FedAA in Algorithm 1 under full participation has*

$$\min_{t \in [T]} \mathbb{E}[\|\nabla f(\theta_t)\|^2] = \mathcal{O}\left(\frac{1}{\sqrt{TKm}}\right).$$

**Remark 6.** *Corollary 1 suggests that with sufficient large $T$, when $T = \mathcal{O}(Km)$, FedNLAA and FedAA achieve a convergence rate of $\mathcal{O}(\frac{1}{T})$. This convergence rate is consistent with the theoretical results established for existing federated non-convex optimization methods, such as FedAMS (Wang et al., 2022b) and FedAdam (Reddi et al., 2020).*

**Partial Participation:** We consider a scenario in which, at each iteration $t$, only $n$ out of $m$ clients engage in local updates and communicate with the central server, such that $|S_t| = n$ for all $t \in [1, T]$. This partial participation inherently involves randomness due to the sampling process, with the associated coefficients depending on the specific sampling strategy employed.

In this study, we focus on random sampling without replacement. Specifically, at the $t$-th iteration, a subset $S_t$ comprising $n$ clients is selected uniformly at random for local updates. For any two clients $i, j \in S_t$, the probability that client $i$ is included in the sampled subset is given by $\mathbb{P}\{i \in S_t\} = \frac{n}{m}$, while the joint probability that both clients $i$ and $j$ are selected simultaneously is $\mathbb{P}\{i, j \in S_t\} = \frac{n(n-1)}{m(m-1)}$.

**Theorem 3.** *Under Assumptions 1-3, if $\eta_l$ satisfies:*

$$\eta_l \leq \min\left\{\frac{1}{8KL}, \frac{n(m-1)\epsilon}{48m(n-1)K_1[3\eta L + C_1^2\eta L + K_2]}\right\},$$

*where $K_1 = K\sqrt{\beta_2 K^2 G^2 + \epsilon}$ and $K_2 = 2\sqrt{2(1-\beta_2)}G$, then FedNLAA in Algorithm 1 under partial participation has*

$$\min_{t\in[T]} \mathbb{E}[\|\nabla f(\theta_t)\|^2] \leq 2K_4\left[\frac{f_0 - f_*}{\eta\eta_l KT} + \frac{\Xi}{T} + \Omega\right],$$

*where $K_4$ is defined in Theorem 1, $\Omega = \mathcal{O}(\frac{K\eta\eta_l + n\eta\eta_l}{n^2}) + \mathcal{O}(\frac{\eta K^3 \eta_l^2}{n}) + \mathcal{O}(K^2\eta_l^2) + \mathcal{O}(\frac{1}{n})$ and $\Xi = \mathcal{O}(K\eta\eta_l)$. The exact expressions of $\Omega$ and $\Xi$ are shown in its proof of Appendix 2.*

**Theorem 4.** *Under Assumptions 1-3, take $\eta_l$ as the one in Theorem 3, then FedAA in Algorithm 1 under partial participation has*

$$\min_{t\in[T]} \mathbb{E}[\|\nabla f(\theta_t)\|^2] \leq 2K_4\left[\frac{f_0 - f_*}{\eta\eta_l KT} + \frac{\Xi}{T} + \Omega\right],$$

*where $K_4$ is defined in Theorem 1, $\Omega = \mathcal{O}(\frac{K\eta\eta_l + n\eta\eta_l}{n^2}) + \mathcal{O}(\frac{\eta K^3 \eta_l^2}{n}) + \mathcal{O}(K^2\eta_l^2) + \mathcal{O}(\frac{1}{n})$ and $\Xi = \mathcal{O}(K\eta\eta_l)$.*

**Remark 7.** *When the parameters $C = D$ and $\frac{C}{\alpha n} = 1$, result of Theorem 3 becomes the one of Theorem 4. The upper bound for $\min_{t\in[T]}\mathbb{E}[\|\nabla f(\theta_t)\|^2]$ contains three terms: The first two terms decrease as $T$ increases, and this term tends to zero as $t$ tends to infinity. The last term involves the local stochastic variance $\sigma_l$ and the global variance $\sigma_g$. Under the i.i.d. assumption, characterized by $\sigma_g = 0$ and identical data distributions across all clients, the variance term $\Omega$ becomes smaller.*

**Corollary 2.** *Suppose choose local learning rate $\eta_l = \Theta(\frac{1}{\sqrt{T}K})$ and global learning rate $\eta = \Theta(\sqrt{Kn})$, the convergence rate for FedNLAA and FedAA in Algorithm 1 under partial participation without replacement sampling is*

$$\min_{t\in[T]}\mathbb{E}[\|\nabla f(\theta_t)\|^2] = \mathcal{O}\left(\frac{\sqrt{K}}{\sqrt{T}n}\right).$$

**Remark 8.** *Corollary 2 indicates that the results in Theorems 3 and 4 are directly influenced by the global variance $\sigma_g^2$. This convergence rate aligns with the partial participation result for FedAvg in the non-i.i.d. setting reported in (Yang et al., 2021). It is observed that the global variance exerts a more pronounced effect on the convergence behavior under partial participation, particularly in highly non-i.i.d. scenarios where $\sigma_g$ is large. The impact of the number of local*

*updates $K$ is nuanced: under partial participation, a larger $K$ leads to slower convergence, whereas the opposite trend occurs under full participation. A similar slowdown has been noted in (Wang et al., 2022b).*

## 4.3  FedNLACA and FedACA Algorithms

In order to reduce communication costs, we introduce two novel methods: **Fed**erated **N**ew **L**azy **A**ggregation **C**ompression **A**MSGrad (FedNLACA) and **Fed**erated **A**ccelerated **C**ompression **A**MSGrad (FedACA). These two algorithms combine two of our proposed strategies (NLA for FedNLACA and AA for FedACA) and compression techniques. The detailed procedure is given in Algorithm 2.

## 4.4  Convergence analysis for FedNLACA and FedACA

For the full participation, we have the following main results.

**Theorem 5.** *Under Assumption 1-4, if the local learning rate $\eta_l$ satisfies:*

$$\eta_l \leq \min\left\{\frac{1}{8KL}, \frac{\epsilon}{KC_{\beta,q}[3\eta L + 2C_2\eta L + K_2]}\right\},$$

*where $K_2$ is defined in Theorem 3 and $C_{\beta,q} = \sqrt{4\beta_2(1+q^2)^3(1-q^2)^{-2}K^2G^2 + \epsilon}$, then the iterates of FedNLACA in Algorithm 2 under the full participation scheme have*

$$\min_{t\in[T]} \mathbb{E}[\|\nabla f(\theta_t)\|^2] \leq 4\sqrt{K_5}\Big[\frac{f_0 - f_*}{\eta\eta_l KT} + \frac{\Xi}{T} + \Omega\Big],$$

*where $K_5 = 4\beta_2\frac{(1+q^2)^3}{(1-q^2)^2}\eta_l^2K^2G^2 + \epsilon$, $\Omega = \mathcal{O}(\frac{\eta\eta_l}{m^3}) + \mathcal{O}(K^2\eta_l^2) + \mathcal{O}(\frac{1}{m})$ and $\Xi = \mathcal{O}(\frac{K\eta\eta_l}{m^2}) + \mathcal{O}(\frac{1}{m})$. The exact expressions of $\Omega$ and $\Xi$ are shown in its proof of Appendix 3.*

**Theorem 6.** *Under Assumption 1-4, take the local learning rate $\eta_l$ as the one in Theorem 5, then the iterates of FedACA in Algorithm 2 under the full participation scheme satisfy*

$$\min_{t\in[T]} \mathbb{E}[\|\nabla f(\theta_t)\|^2] \leq 4\sqrt{K_5}\Big[\frac{f_0 - f_*}{\eta\eta_l KT} + \frac{\Xi}{T} + \Omega\Big],$$

*where $K_5$ is defined in Theorem 5, $\Omega = \mathcal{O}(\frac{\eta\eta_l}{m^3}) + \mathcal{O}(K^2\eta_l^2) + \mathcal{O}(\frac{1}{m})$ and $\Xi = \mathcal{O}(\frac{K\eta\eta_l}{m^2}) + \mathcal{O}(\frac{1}{m})$.*

**Remark 9.** *When the parameters $C = D, \frac{C}{\alpha m} = 1$, the result of Theorem 5 becomes the result of Theorem 6. The upper bound for $\min_{t\in[T]} \mathbb{E}[\|\nabla f(\theta_t)\|^2]$ consists of three terms. The first two terms decay as $T$ increases and vanish as $t$ tends to infinity. The last term involves both the local stochastic variance $\sigma_l$ and the global variance $\sigma_g$, which captures the degree of data heterogeneity. In the i.i.d. setting where all clients share the same data distribution, i.e., $\sigma_g = 0$, the variance term $\Omega$ becomes smaller.*

**Corollary 3.** *Suppose we choose local learning rate $\eta_l = \Theta(\frac{1}{\sqrt{T}K})$ and the global learning rate $\eta = \Theta(\sqrt{Km})$, when $T$ is sufficiently large, i.e., $T \geq Km$, the convergence rate for FedNLACA*

---

**Algorithm 2:** FedNLACA and FedACA

---

**Input:** initial value $\theta_1$, local step size $\eta_l$, global step size $\eta$, constant $\beta_1, \beta_2, \epsilon$, for each client $i \in S_t$, $\Delta_0^i = 0$, compressor $C(\cdot)$

**Initialization:** $\mathbf{m}_0 \leftarrow 0, \mathbf{v}_0 \leftarrow 0, \mathbf{e}_1^i = 0$.

**for** $t = 1$ **to** $T$ **do**

    Randomly select a subset of clients $S_t$ and the server transmits $\theta_t$ to the subset of clients $S_t$.

    $\theta_{t,0}^i = \theta_t$.

    **for** *each client $i \in S_t$ in parallel* **do**

        **for** $k = 0, ..., K - 1$ **do**

            • Compute local stochastic gradient: $\mathbf{g}_{t,k}^i = \nabla F_i(\theta_{t,k}^i; \xi_{t,k}^i)$

            • $\theta_{t,k+1}^i = \theta_{t,k}^i - \eta_l \mathbf{g}_{t,k}^i$

        **end**

        $\Delta_t^i = \theta_{t,K}^i - \theta_t$

        • Compress $\widehat{\Delta}_t^i = \mathcal{C}(\Delta_t^i + \mathbf{e}_t^i)$, $\mathcal{C}(q_t^i) = \widehat{\Delta}_t^i - \widehat{\Delta}_{t-1}^i$,

        • Judge: If $\mathcal{C}(q_t^i)$ satisfies NLA (Example 2) or AA (Example 5),

        • then outputs the result $\widehat{\widehat{\Delta}}_t^i$ of the judgement and passes it to the server and update $\mathbf{e}_{t+1}^i = \Delta_t^i + \mathbf{e}_t^i - \widehat{\widehat{\Delta}}_t^i$

    **end**

    **for** *each client $j \notin S_t$ in parallel do* **do**

        • client $j$ preserves the outdated compression error by setting $\mathbf{e}_{t+1}^j = \mathbf{e}_t^j$.

    **end**

    •The server consolidates the local updates: $\widehat{\Delta}_t = \frac{1}{|S_t|} \sum_{i \in \mathcal{S}_t} \widehat{\widehat{\Delta}}_t^i$

    • The server updates the parameter vector $\mathbf{x}_{t+1}$ by utilizing $\hat{\Delta}_t$ following the procedure outlined in Algorithm 1. (Line 15-18)

**end**

---

and FedACA in Algorithm 2 under full participation scheme satisfies

$$\min_{t \in [T]} \mathbb{E}[\|\nabla f(\theta_t)\|^2] = \mathcal{O}\left(\frac{1}{\sqrt{TKm}}\right).$$

**Remark 10.** *Corollary 3 suggests that with sufficient large $T$, FedNLACA and FedACA achieve the desired $\mathcal{O}(\frac{1}{\sqrt{TKm}})$ convergence rate which matches the result for its uncompressed counterpart FedNLAA and FedAA. In addition, when $T = \mathcal{O}(Km)$, $\min_{t \in [T]} \mathbb{E}[\|\nabla f(\theta_t)\|^2] = \mathcal{O}\left(\frac{1}{T}\right)$. This suggests that FedNLACA and FedACA can indeed achieve better communication efficiency without sacrificing much on the accuracy.*

For the partial participation, we have the following main results.

**Theorem 7.** *Under Assumption 1-4, take the local learning rate $\eta_l$ as the one in Theorem 5, then the iterates of FedNLACA in Algorithm 2 under partial participation scheme have*

$$\min_{t \in [T]} \mathbb{E}[\|\nabla f(\theta_t)\|^2] \leq 8\sqrt{K_5} \left[ \frac{f_0 - f_*}{\eta \eta_l KT} + \frac{\Xi}{T} + \Omega \right],$$

*where $K_5$ is defined in Theorem 5, $\Omega = \mathcal{O}(\frac{K\eta\eta_l + n\eta\eta_l}{n^2}) + \mathcal{O}(\frac{\eta K^3 \eta_l^2}{n}) + \mathcal{O}(K^2 \eta_l^2) + \mathcal{O}(\frac{1}{n})$ and $\Xi = \frac{1}{n} + \mathcal{O}(\frac{K\eta\eta_l}{n^2})$. The exact expressions of $\Omega$ and $\Xi$ are shown in its proof of Appendix 3.*

**Theorem 8.** *Under Assumption 1-4, take the local learning rate $\eta_l$ as the one in Theorem 5, then the iterates of FedACA in Algorithm 2 under partial participation scheme satisfy*

$$\min_{t \in [T]} \mathbb{E}[\|\nabla f(\theta_t)\|^2] \leq 8\sqrt{K_5} \left[ \frac{f_0 - f_*}{\eta \eta_l KT} + \frac{\Xi}{T} + \Omega \right],$$

*where $K_5$ is defined in Theorem 5, $\Omega = \mathcal{O}(\frac{K\eta\eta_l + n\eta\eta_l}{n^2}) + \mathcal{O}(\frac{\eta K^3 \eta_l^2}{n}) + \mathcal{O}(K^2 \eta_l^2) + \mathcal{O}(\frac{1}{n})$ and $\Xi = \frac{1}{n} + \mathcal{O}(\frac{K\eta\eta_l}{n^2})$.*

**Remark 11.** *When the parameters $C = D, \frac{C}{\alpha n} = 1$, the result of Theorem 7 becomes the result of Theorem 8. The upper bound for $\min_{t \in [T]} \mathbb{E}[\|\nabla f(\theta_t)\|^2]$ contains three terms: The first two terms decrease as $T$ increases, and this term tends to zero as $t$ tends to infinity. The final term is associated with the local stochastic variance $\sigma_l$ and the global variance $\sigma_g$. Under the i.i.d. assumption, characterized by $\sigma_g = 0$ and identical data distributions across clients, the variance term $\Omega$ becomes smaller.*

**Corollary 4.** *Suppose we choose local learning rate $\eta_l = \Theta(\frac{1}{\sqrt{T}K})$ and the global learning rate $\eta = \Theta(\sqrt{Kn})$, the convergence rate for FedNLACA, FedACA in Algorithm 2 under partial participation scheme without replacement sampling is*

$$\min_{t \in [T]} \mathbb{E}[\|\nabla f(\theta_t)\|^2] = \mathcal{O}\left(\frac{\sqrt{K}}{\sqrt{Tn}}\right).$$

**Remark 12.** *Corollary 4 indicates that the results in Theorems 7 and 8 are directly influenced by the global variance $\sigma_g^2$. This convergence rate aligns with the partial participation result for FedAvg in the non-i.i.d. setting reported in (Yang et al., 2021). It is observed that the global*

*variance exerts a more pronounced effect on the convergence behavior under partial participation, particularly in highly non-i.i.d. scenarios where $\sigma_g$ is large. The impact of the number of local updates $K$ is nuanced: under partial participation, larger values of $K$ lead to slower convergence, whereas the opposite trend occurs under full participation. A similar slowdown has been noted in (Wang et al., 2022b).*

## 4.5 FedBNLACA and FedBACA Algorithms

---

**Algorithm 3:** FedBNLACA and FedBACA.

---

**Input:** initial value $\theta_1, \theta_0 = 0$, local step size $\eta_l$, global step size $\eta$, constants $\beta_1$, $\beta_2$ and $\epsilon$, for each client $i \in S_t$, $\Delta_0^i = 0$, compressor $C(\cdot)$.

**Initialization:** $\mathbf{m}_0 \leftarrow 0, \mathbf{v}_0 \leftarrow 0, \mathbf{e}_1^i = 0, \mathbf{E}_1^i = 0..$

**for** $t = 0, 1, 2, \cdots, T-1$ **do**

    *On the server:* Server randomly selects a subset of clients $S_t$.

    **for** *each client $i \in S_t$ in parallel* **do**
- $\theta_t^i = \theta_t$
- Compress: $\widehat{\theta_t^i} = \mathcal{C}(\theta_t^i + \mathbf{E}_t^i), \mathcal{C}(Q_t^i) = \widehat{\theta_t^i} - \widehat{\theta_{t-1}^i}$,
- Judge: If $\mathcal{C}(Q_t^i)$ satisfies NLA ( Example 3) or AA (Example 6), output: $\widehat{\widetilde{\theta}}_t^i$
- Update: $\mathbf{E}_{t+1}^i = \theta_t^i + \mathbf{E}_t^i - \widehat{\widetilde{\theta}}_t^i$,

    **end**

    **for** *each client $j \notin S_t$ in parallel* **do**
- maintain stale compression error $\mathbf{E}_{t+1}^j = \mathbf{E}_t^j$.

    **end**

    On the clients: $\theta_{t,0}^i = \widehat{\widetilde{\theta}}_t^i$ **for** *each client $i \in S_t$ in parallel* **do**

        **for** $k = 0, ..., K-1$ **do**
- Compute local SGD: $\mathbf{g}_{t,k}^i = \nabla F_i(\theta_{t,k}^i; \xi_{t,k}^i)$,
$\theta_{t,k+1}^i = \theta_{t,k}^i - \eta_l \mathbf{g}_{t,k}^i.$

        **end**
- $\Delta_t^i = \theta_{t,K}^i - \widehat{\widetilde{\theta}}_t^i$
- As in Algorithm 2 (Line 12-14)

    **end**

    As in Algorithm 2 (Line 16-20)

**end**

---

Algorithm 2 gives only a one-way algorithm to reduce the cost of communication (uplink). In the section, we propose two bidirectional communication algorithms (uplink and downlink) with efficiently adaptive non-convex optimization: **Fed**erated **B**idirectional **N**ew **L**azy **A**ggregation **C**ompression **A**MSGrad (FedBNLACA) and **Fed**erated **B**idirectional **A**ccelerated **C**ompression **A**MSGrad (FedBACA). The detailed procedure is given in Algorithm 3.

Subsequently, we present the convergence analysis for FedBNLACA and FedBACA. Owing to space constraints, the analysis is provided exclusively for the full participation scenario, while the partial participation case is addressed in Appendix 4.

**Theorem 9.** *Under Assumptions 1-4, take the local learning rate $\eta_l$ as the one in Theorem 5,*

*then FedBNLACA in Algorithm 3 under the full participation has*

$$\min_{t \in [T]} \mathbb{E}[\|\nabla f(\theta_t)\|^2] \leq 4\sqrt{K_5}\Big[\frac{f_0 - f_*}{\eta \eta_l KT} + \frac{\Xi}{T} + \Omega\Big],$$

*where $K_5$ is defined in Theorem 5, $\Omega = \mathcal{O}(\frac{\eta \eta_l}{m^3}) + \mathcal{O}(\frac{K^2 \eta^2 \eta_l^2}{m^2}) + \mathcal{O}(\frac{\eta^2}{m}) + \mathcal{O}(\frac{\eta^2 \eta K}{m})$ and $\Xi = \mathcal{O}(\frac{K \eta \eta_l}{m^2}) + \mathcal{O}(\frac{1}{m})$. The exact expressions of $\Omega$ and $\Xi$ are shown in its proof of Appendix 4.*

**Theorem 10.** *Under Assumptions 1-4, take the local learning rate $\eta_l$ as the one in Theorem 5, then FedBACA in Algorithm 3 under the full participation has*

$$\min_{t \in [T]} \mathbb{E}[\|\nabla f(\theta_t)\|^2] \leq 4\sqrt{K_5}\Big[\frac{f_0 - f_*}{\eta \eta_l KT} + \frac{\Xi}{T} + \Omega\Big],$$

*where $K_5$ is defined in Theorem 5, $\Omega = \mathcal{O}(\frac{\eta \eta_l}{m^3}) + \mathcal{O}(\frac{K^2 \eta^2 \eta_l^2}{m^2}) + \mathcal{O}(\frac{\eta^2}{m}) + \mathcal{O}(\frac{\eta^2 \eta K}{m})$ and $\Xi = \mathcal{O}(\frac{K \eta \eta_l}{m^2}) + \mathcal{O}(\frac{1}{m})$.*

**Remark 13.** *When the parameters $C = D$ and $\frac{C}{\alpha m} = 1$, result of Theorem 9 becomes the one of Theorem 10. The upper bound for $min_{t \in [T]} \mathbb{E}[\|\nabla f(\theta_t)\|^2]$ contains three terms: The first two terms decrease as $T$ increases, and this term tends to zero as $t$ tends to infinity. The final term is associated with the local stochastic variance $\sigma_l$ and the global variance $\sigma_g$. Under the i.i.d. assumption, characterized by $\sigma_g = 0$ and identical data distributions across clients, the variance term $\Omega$ becomes smaller.*

**Corollary 5.** *Suppose choose local learning rate $\eta_l = \Theta(\frac{1}{\sqrt{TK}})$ and global learning rate $\eta = \Theta(\frac{1}{\sqrt{T}})$, when $T$ is sufficiently large, i.e., $T = \mathcal{O}(Km)$, the convergence rate for FedBNLACA and FedBACA in Algorithm 3 under full participation has*

$$\min_{t \in [T]} \mathbb{E}[\|\nabla f(\theta_t)\|^2] = \mathcal{O}\Big(\frac{1}{T}\Big).$$

**Remark 14.** *Corollary 5 shows that when $T$ is sufficiently large, both FedBNLACA and Fed-BACA achieve a convergence rate of $\mathcal{O}(1/T)$. This rate is consistent with the theoretical results reported for existing federated non-convex optimization algorithms, including FedAMS (Wang et al., 2022b) and FedAdam (Reddi et al., 2020).*

## 5 Experiments

We compare our proposed algorithms with several state-of-the-art baselines, including FedAvg (McMahan et al., 2017), FedAMS (Wang et al., 2022b), FedAdam (Reddi et al., 2020), LENA (Ghadikolaei et al., 2021), FedPAQ (Reisizadeh et al., 2020), and two LAG-type baselines, denoted as LAG-FedAMS($R = 1$) and LAG-FedAMS($R = 5$), corresponding to single-step and multi-step lazy aggregation rules, respectively. We use MNIST and Fashion-MNIST datasets, and MLP and CNN models, respectively. A total of 100 clients for all federated training experiments are used. Set the partial participation rate to 0.5, i.e. in each round, the server selects 50 clients out of 100 to participate in communication and model updating. In each round, the client completes 3 local epochs with batch size 32. In addition to the MNIST and Fashion-MNIST experiments,

we further conduct a scalability evaluation on CIFAR-10 with a VGG-11 model. Compared with the previous two datasets and smaller neural network models, CIFAR-10 contains more complex natural images and VGG-11 has a deeper convolutional architecture, making this setting more challenging for federated optimization. In this experiment, we use 40 clients with a partial participation rate of 0.2, so that 8 clients are selected in each communication round. For all main-text experiments reported in this section, each method is repeated over three independent random seeds. The reported curves show the mean performance across the three runs, with shaded regions indicating one standard deviation. For tabular results, we report the mean and standard deviation across the three random seeds. In experiments, we respectively sample Independent Identical Distribution (I.I.D.) and non-I.I.D. client data from the dataset. Choose compression rate in Top$k$ to be 1/8 and 1/128. For parameters $C$ and $D$, the values are not too large. From theoretical analysis, the larger the values of $C$ and $D$, the larger the errors. In addition, $C$ and $D$ are too small, and basically does not help to reduce communication cost. We also verify this result. We suggest that $C$ and $D$ in the vicinity of $\frac{\sqrt{\alpha \lg m}}{m}$.

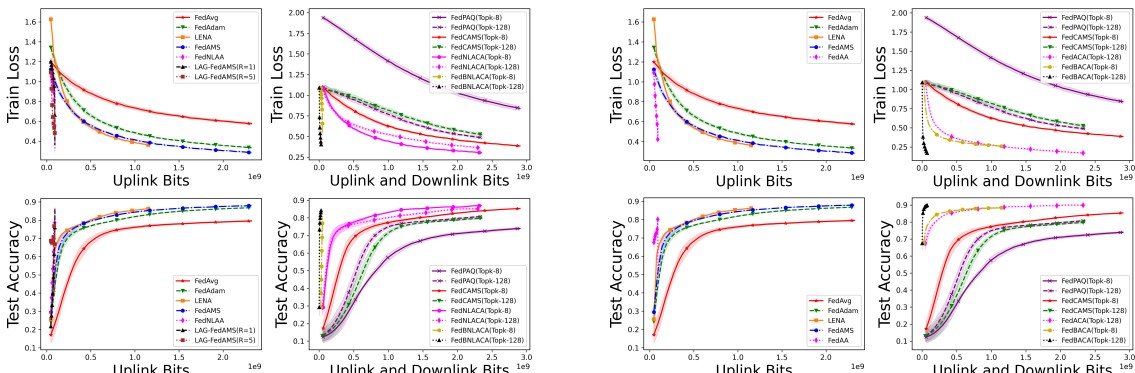

Figure 1: Algorithm comparison: NLA strategy, on Fashion-MNIST dataset via CNN model, and the I.I.D. client data sampling.

Figure 2: Algorithm comparison: AA strategy, on the Fashion-MNIST dataset via CNN model, and the I.I.D. client data sampling.

Figures 1-2 represent the relationship between the accuracy of prediction and communication Bits when the model is CNN and the I.I.D. client data is sampled from Fashion-MNIST dataset. From Figure 1, we can find that (i) the proposed algorithm FedNLAA not only communicates fewer bits than the other three state-of-the-art algorithms, but also has higher accuracy; (ii) our proposed algorithms (FedNLACA and FedBNLACA) require only few communication bits to achieve good accuracy, especially the bidirectional compression algorithm FedBLACA requires even fewer Bits. These shows that our proposed algorithms are communication efficient. From Figure 2, we can find that (i) the proposed algorithm FedAA can converge more quickly than the other three algorithms, thus disguising the reduction of communication cost; (ii) FedACA and FedBACA can also converge quickly and with higher accuracy than the other three algorithms, especially FedBACA.

Figures 3–4 report the relationship between training loss, test accuracy, and communication cost on CIFAR-10 with VGG-11 under the non-I.I.D. federated setting. All curves are averaged over three independent random seeds, and the shaded regions indicate one standard deviation. Compared with the MNIST and Fashion-MNIST experiments, this setting is more challenging because CIFAR-10 has a more complex image distribution and VGG-11 is a deeper convolutional

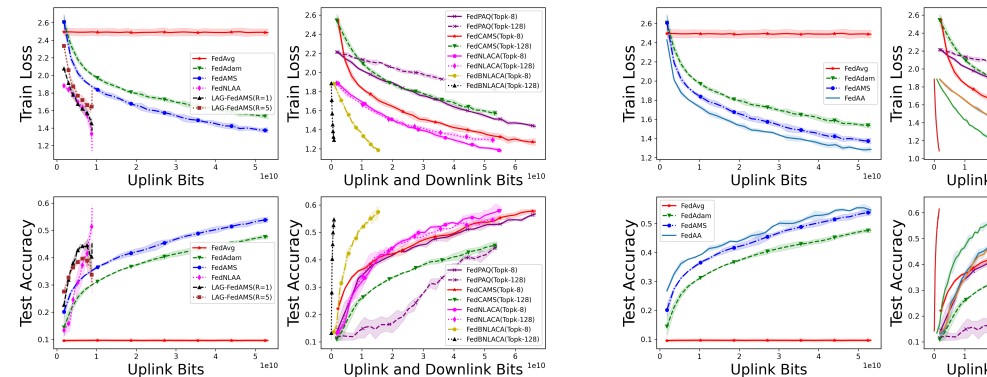

Figure 3: Algorithm comparison: NLA strategy, on CIFAR10 dataset via VGG11 model, and the no-I.I.D. client data sampling.

Figure 4: Algorithm comparison: AA strategy, on CIFAR10 dataset via VGG11 model, and the no-I.I.D. client data sampling.

model.

From Figure 3, we observe that FedAvg performs poorly on CIFAR-10, whereas adaptive optimization methods such as FedAdam and FedAMS achieve better convergence and test accuracy. The proposed NLA-based methods further improve the communication–accuracy trade-off. In particular, FedNLAA achieves rapid early-stage improvement with reduced uplink communication, while the LAG-FedAMS baselines reduce communication at the cost of lower final accuracy. When Top-$k$ compression and error feedback are incorporated, FedNLACA and FedBNLACA achieve competitive accuracy under a smaller communication budget. The bidirectional variant FedBNLACA is especially communication-efficient because it reduces both uplink and downlink communication while maintaining favorable test accuracy.

From Figure 4, we further examine the AA-based variants. FedAA converges faster than the standard adaptive baselines, suggesting that the accelerated aggregation mechanism improves communication efficiency in the non-I.I.D. setting. The compressed AA variants, including FedACA and FedBACA, also achieve favorable accuracy with reduced communication cost. In particular, FedBACA provides a strong communication–accuracy trade-off by combining Top-$k$ compression, error feedback, and bidirectional communication reduction. These results demonstrate that the proposed NLA/AA mechanisms remain effective on CIFAR-10 with VGG-11, providing additional evidence for scalability beyond relatively simple datasets and small models.

Table 1 reports a component-wise ablation and communication diagnostic study on CIFAR-10 with VGG-11 under the I.I.D. federated setting. The table separates the effects of Top-$k$ compression, error feedback, NLA/AA-type adaptive communication, LAG-type lazy aggregation, and bidirectional communication reduction. The column "Bits to Target" reports the total communication cost required to first reach 15% test accuracy, while "Final Total Bits" reports the total communication cost after training. All communication costs are measured in Gbits.

The results show that standard adaptive optimizers, such as FedAdam and FedAMS, substantially improve over FedAvg but incur the largest final communication cost. LAG-FedAMS reduces communication, but its final accuracy is lower, especially when $R = 5$. Compression-only methods, such as FedPAQ and FedCAMS, reduce communication by decreasing the uplink message size, but their performance is sensitive to the compression level. In contrast, the proposed NLA/AA-based variants provide a better communication–accuracy trade-off. In par-

Table 1: Component-wise ablation and communication diagnostic results on CIFAR-10 with VGG-11 under the I.I.D. federated setting. "Bidir." indicates whether bidirectional communication reduction is used. "Bits to Target" denotes the total communication cost required to first reach 15% test accuracy. Communication costs are reported in Gbits ($10^9$ bits). All results are averaged over three independent random seeds and reported as mean $\pm$ standard deviation.

| Method | Top-$k$ | EF | Strategy | Bidir. | Lag $R$ | Final Acc. (%) | Bits to Target | Final Total Bits |
|---|---|---|---|---|---|---|---|---|
| FedAvg | $\times$ | $\times$ | None | $\times$ | – | $9.65 \pm 0.46$ | Not reached | $106.34 \pm 0.00$ |
| FedAdam | $\times$ | $\times$ | None | $\times$ | – | $47.69 \pm 0.84$ | $5.91 \pm 1.67$ | $106.34 \pm 0.00$ |
| FedAMS | $\times$ | $\times$ | None | $\times$ | – | $53.96 \pm 1.00$ | $3.54 \pm 0.00$ | $106.34 \pm 0.00$ |
| LAG-FedAMS($R = 1$) | $\times$ | $\times$ | LAG | $\times$ | 1 | $45.90 \pm 0.52$ | $3.54 \pm 0.00$ | $62.03 \pm 0.00$ |
| LAG-FedAMS($R = 5$) | $\times$ | $\times$ | LAG | $\times$ | 5 | $29.85 \pm 2.92$ | $3.54 \pm 0.00$ | $62.03 \pm 0.00$ |
| FedNLAA | $\times$ | $\times$ | NLA | $\times$ | – | $58.24 \pm 1.51$ | $6.50 \pm 2.30$ | $62.03 \pm 0.00$ |
| FedAA | $\times$ | $\times$ | AA | $\times$ | – | $54.71 \pm 1.03$ | $3.54 \pm 0.00$ | $106.34 \pm 0.00$ |
| FedPAQ(Topk-8) | $\checkmark$ | $\times$ | None | $\times$ | – | $56.70 \pm 0.88$ | $2.95 \pm 1.04$ | $66.46 \pm 0.00$ |
| FedPAQ(Topk-128) | $\checkmark$ | $\times$ | None | $\times$ | – | $44.80 \pm 0.55$ | $13.20 \pm 4.72$ | $54.00 \pm 0.00$ |
| FedCAMS(Topk-8) | $\checkmark$ | $\checkmark$ | None | $\times$ | – | $57.79 \pm 0.72$ | $2.22 \pm 0.00$ | $66.46 \pm 0.00$ |
| FedCAMS(Topk-128) | $\checkmark$ | $\checkmark$ | None | $\times$ | – | $45.23 \pm 1.44$ | $4.80 \pm 0.85$ | $54.00 \pm 0.00$ |
| FedNLACA(Topk-8) | $\checkmark$ | $\checkmark$ | NLA | $\times$ | – | $58.23 \pm 0.93$ | $3.64 \pm 1.01$ | $55.39 \pm 0.00$ |
| FedNLACA(Topk-128) | $\checkmark$ | $\checkmark$ | NLA | $\times$ | – | $54.52 \pm 1.20$ | $4.19 \pm 0.84$ | $53.31 \pm 0.00$ |
| FedBNLACA(Topk-8) | $\checkmark$ | $\checkmark$ | NLA | $\checkmark$ | – | $57.86 \pm 0.92$ | $1.35 \pm 0.66$ | $15.51 \pm 0.00$ |
| FedBNLACA(Topk-128) | $\checkmark$ | $\checkmark$ | NLA | $\checkmark$ | – | $54.86 \pm 0.77$ | $0.09 \pm 0.02$ | $0.97 \pm 0.00$ |
| FedACA(Topk-8) | $\checkmark$ | $\checkmark$ | AA | $\times$ | – | $60.41 \pm 1.44$ | $4.20 \pm 1.70$ | $54.00 \pm 0.00$ |
| FedACA(Topk-128) | $\checkmark$ | $\checkmark$ | AA | $\times$ | – | $61.65 \pm 1.36$ | $3.60 \pm 1.47$ | $54.00 \pm 0.00$ |
| FedBACA(Topk-8) | $\checkmark$ | $\checkmark$ | AA | $\checkmark$ | – | $60.29 \pm 1.26$ | $1.77 \pm 0.72$ | $26.59 \pm 0.00$ |
| FedBACA(Topk-128) | $\checkmark$ | $\checkmark$ | AA | $\checkmark$ | – | $61.43 \pm 1.62$ | $0.09 \pm 0.05$ | $1.66 \pm 0.00$ |

ticular, FedNLACA and FedACA improve upon compression-only baselines by incorporating adaptive communication, while FedBNLACA and FedBACA further reduce the total communication cost through bidirectional communication reduction. These results indicate that the observed communication savings are not due to compression alone, but also come from the proposed NLA/AA strategies and their bidirectional extensions.

## 6 Conclusion

We propose two novel strategies: NLA and AA in the framework of federated learning. They are simple to operate and effective in reducing the communication cost. The NLA strategy achieves communication cost reduction by reducing the amount of information passed and AA strategy reduces the communication cost by accelerating computation. By combining our proposed strategies with compression techniques, we design FedNLAA and FedAA algorithms, which not only achieve communication cost reduction in one-way, but also extend them to bidirectional algorithms (FedBNLACA and FedBACA), which achieve communication cost reduction in bidirectional communication as well.

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
