# OpenReview forum: "Strategy-driven Bidirectional Efficient Adaptive Federated Learning"
_SLADS/Section_C — Decision pending for SLADS_Section_C_

### Review · Reviewer_8iDi · 2026-04-26

**Summary Of Contributions:**

This paper studies communication-efficient adaptive federated learning under non-IID data, partial client participation, and costly uplink/downlink communication. It proposes strategy-driven aggregation mechanisms, mainly New Lazy Aggregation (NLA) and Accelerated Aggregation (AA), and integrates them with adaptive FL methods such as FedAMS/FedCAMS.The resulting algorithms aim to reduce communication by selectively skipping or compressing client-server transmissions while preserving convergence guarantees. The supplementary material provides non-convex convergence proofs for full and partial participation settings, including variants with compression and bidirectional communication reduction.Experiments on MNIST and Fashion-MNIST suggest improved communication efficiency over FedAvg, FedAMS, and FedAdam

**Audience:**

Yes

**Broader Impact Concerns:**

I do not identify any major ethical concern in this work. The paper focuses on communication-efficient federated learning, which may have positive effects on privacy-preserving and resource-efficient training.

**Claims And Evidence:**

Yes

**Requested Changes:**

## Major Concerns

1. **Limited theoretical evidence for communication efficiency.**
   The paper is mainly motivated by reducing communication, but the current theory is still mostly a convergence analysis, stated in terms of
   $
   \min_{t\in[T]} \mathbb{E}\|\nabla f(\theta_t)\|^2 .
   $
   It does not say much about the communication side itself. For example, there is no bound on the total number of transmitted bits, nor on how often the NLA/AA conditions are expected to skip uploads or downloads. Because of this, the theory shows that the proposed methods converge, but it does not really prove that they save communication. Some analysis of the expected communication cost induced by the NLA/AA rules would make the main claim much more convincing.

2. **The baselines may not be sufficiently strong or comprehensive.**
   The reported comparisons are mainly with standard FL optimizers such as FedAvg, FedAMS, and FedAdam. These are useful references, but they are not enough to demonstrate the advantage of a communication-efficient method. Since the paper is positioned around compression, lazy aggregation, and bidirectional communication reduction, it would be more convincing to include stronger baselines from these directions, such as methods using error feedback, compressed updates, or lazy communication.

3. **Communication savings are not separated clearly from overall performance.**
   Although the figures plot performance against communication bits, it is still hard to tell how much of the gain comes from the proposed NLA/AA rules themselves. The paper would benefit from more direct measurements, such as the skipped-transmission ratio, separate uplink and downlink costs, and total bits used to reach a fixed accuracy. These results would make the communication-saving mechanism clearer, rather than only showing that the final curves look better.

## Minor Concerns

1. **Please double-check the algorithm names and theorem references.**
   There are many similarly named variants in the paper, which makes consistency especially important. For example, in Theorem 3, it seems that the algorithm should be FedNLAA rather than FedLAA. It would be good to go through the paper and make sure the names used in the algorithms, theorems, remarks, and experiments are consistent.

2. **The manuscript still needs careful proofreading.**
   There are a number of grammatical and typographical issues, such as “distributiona common characteristic,” “th one,” “unplink,” and “some postive constants.” These are not major technical problems, but they do make the paper harder to read.

3. **Some notation appears inconsistent.**
   For instance, Theorem 5 says that the local learning rate $\eta_l$ should satisfy a certain condition, but the displayed condition starts with $\eta \leq \ldots$ instead. Similar small notation issues appear in a few other convergence statements, so the theoretical section would benefit from a careful notation check.

4. **A small ablation on C and D would be useful.**
   The paper gives some guidance on choosing $C$ and $D$, and also notes that values that are too large or too small can hurt performance. However, this is not clearly supported by experiments. A simple sensitivity study over $C$ and $D$ would make the practical recommendation more convincing.

**Strengths And Weaknesses:**

The paper addresses an important problem in federated learning: reducing communication costs for adaptive optimization under non-IID data and partial client participation. Its main strength is the unified integration of strategy-driven aggregation, compression, error feedback, and bidirectional communication reduction, supported by convergence analysis and empirical results showing improved communication efficiency over standard baselines.

However, the paper includes many algorithmic variants, which makes the central novelty somewhat difficult to identify. The theory mainly provides standard first-order stationarity guarantees, but does not directly quantify communication complexity, such as total transmitted bits or expected skipped transmissions needed to reach a target accuracy. The experiments are also relatively limited, relying mainly on MNIST/Fashion-MNIST and standard baselines; larger datasets and comparisons with more recent communication-efficient FL methods would make the results more convincing.

Overall, the paper is relevant and technically useful, but its contribution would be stronger with clearer positioning of each proposed strategy and more direct theoretical and empirical evidence for communication savings.

---

> ### Author Response · Authors · 2026-06-10
>
> ## **Major Concerns 1**
>
> We thank the reviewer for this important comment. We agree that the original
> theoretical results mainly established convergence in terms of
> $\min_{t\in[T]}\mathbb{E}\|\nabla f(\theta_t)\|^2$, but did not explicitly
> quantify the communication cost induced by the proposed NLA/AA rules. In the
> revised manuscript, we have added a communication-cost analysis that accounts for
> the total number of transmitted bits and the expected triggering frequency of the
> proposed strategies.
>
>
>
> ## *Theoretical communication accounting*
> **1. Expected communication cost induced by NLA.**
>
> We first introduce communication indicators for the NLA rule. Let
> $I_{i,t}^{u}=1$ denote that client $i$'s uplink transmission is skipped at
> round $t$ by NLA, and let $I_{i,t}^{d}=1$ denote that the corresponding
> downlink transmission is skipped in the bidirectional setting. Let $b_u$ and
> $b_d$ be the number of bits required to transmit one uplink and one downlink
> object, respectively. Let $b_0\ll b_u$ denote the small number of control bits
> used to indicate that the previous object is reused. For a Top-$k$ compressor,
> for example, the uplink message size can be written as
>
> $$
>     b_u(k)
>     =
>     k\{b_{\rm val}+\lceil \log_2 d\rceil\}+b_{\rm meta},
> $$
>
> where $b_{\rm val}$ is the number of bits used to encode each transmitted
> value, $[\log_2 d]$ bits are used to encode each selected
> coordinate, and $b_{\rm meta}$ denotes the additional metadata.
>
> Under this notation, the total uplink communication cost of NLA up to round
> $T$ is
>
> $$
> B_{\rm NLA}^{u}(T)=\sum_{t=1}^{T}\sum_{i\in S_t}\left[(1-I_{i,t}^{u})b_u+I_{i,t}^{u}b_0\right].$$
>
> Taking expectation gives
>
>    $$
>  \mathbb{E}B_{\rm NLA}^{u}(T)=\sum_{t=1}^{T}\sum_{i=1}^{m}\mathbb{P}(i\in S_t)[b_u-(b_u-b_0)\mathbb{P}(I_{i,t}^{u}=1\mid i\in S_t)].$$
>
> Therefore, compared with the non-skipping baseline, the expected number of saved
> uplink bits is
>     $$(b_u-b_0)
>     \sum_{t=1}^{T}\sum_{i=1}^{m}\mathbb{P}(i \in S_t)\mathbb{P}(I_{i,t}^{u}=1\mid i\in S_t)$$
>
> An analogous expression is added for the downlink communication cost in the
> bidirectional algorithms by replacing $I_{i,t}^{u}$ and $b_u$ with
> $I_{i,t}^{d}$ and $b_d$.
>
> We further provide a sufficient condition for nontrivial expected communication
> saving. Define the relative variation ratio
>
> $$
>     R_{i,t}^{u} = \frac{\|x_{i,t}^{u}-x_{i,t-1}^{u}\|} {\|x_{i,t-1}^{u}\|+\varepsilon},
> $$
>
> where $x_{i,t}^{u}4 denotes the uplink object to be transmitted, such as a
> local update or a compressed update, and $\varepsilon>0$ is used to avoid
> division by zero. The NLA rule skips the uplink transmission when
>
> $$
>     R_{i,t}^{u}\leq \tau_t,
>     \qquad
>     \tau_t=\frac{C}{\alpha |S_t|}.
> $$
>
> Suppose that
>
> $$
>     \mathbb{E}\{(R_{i,t}^{u})^2\mid i\in S_t\}\leq \nu_{i,t}^2.
> $$
>
> Then, by Markov's inequality,
>
> $$
>     \mathbb{P}(R_{i,t}^{u}>\tau_t\mid i\in S_t)
>     \leq
>     \frac{\nu_{i,t}^2}{\tau_t^2}.
> $$
>
> Equivalently,
>
> $$
>     \mathbb{P}(I_{i,t}^{u}=1\mid i\in S_t)
>     =
>     \mathbb{P}(R_{i,t}^{u}\leq \tau_t\mid i\in S_t)
>     \geq
>     \left(1-\frac{\nu_{i,t}^2}{\tau_t^2}\right)_+ .
> $$
>
> Consequently, the expected uplink communication cost admits the upper bound
>
> $$
>     \mathbb{E}B_{\rm NLA}^{u}(T)\leq\sum_{t=1}^{T}\sum_{i=1}^{m} \mathbb{P}(i\in S_t)
>   [b_0+(b_u-b_0)\min(1,\frac{\nu_{i,t}^2}{\tau_t^2}) ].
> $$
>
> This bound shows that NLA yields a nontrivial expected communication saving when
> the transmitted update trajectory is relatively stable, namely when
> $\nu_{i,t}^2\ll \tau_t^2$.
>
> **2. Communication metric for AA.**
>
> For AA, we clarify that it is not a pure per-round skipping rule, but a
> conditional accumulation strategy. Therefore, its communication efficiency should
> not be measured only by the number of skipped messages in a single round. Instead,
> we quantify its communication effect through the total number of bits needed to
> reach a target accuracy. For a target $\epsilon>0$, define
>
> $$T_\epsilon=\inf(t:\min_{s\leq t}\mathbb{E}\|\nabla f(\theta_s)\|^2\leq\epsilon),   B(\epsilon)=\mathbb{E}B(T_\epsilon),
> $$
>
> where $B(T_\epsilon)$ denotes the total communication cost up to the first
> round at which the target stationarity level is reached. This metric captures
> the possible communication benefit of AA through reducing the number of
> communication rounds required to reach a target accuracy.
>
> **Therefore, the revised manuscript now includes theoretical communication cost calculations and theoretical analysis. Specifically, Section 5 reports in detail the experimental results for uplink, downlink, and total communication costs, as well as other communication-related metrics. The corresponding theoretical communication analysis is provided in Appendix 1.4. These new additions are presented in the sections and tables highlighted in blue.**

---

> > ### Author Response · Authors · 2026-06-10
> >
> > ## **Major Concerns 2-3**
> >
> > We thank the reviewer for these constructive comments. We agree that the original experiments mainly compared with standard FL optimizers, such as FedAvg, FedAdam, and FedAMS, and did not sufficiently isolate the sources of communication savings. Since our work focuses on compression, lazy/adaptive aggregation, and bidirectional communication reduction, we have expanded the experimental section accordingly. The main revisions are as follows.
> >
> > **Larger dataset and model.**
> >     We have added new experiments on CIFAR-10 with a VGG-11 model under the non-IID federated setting. This setting is more challenging than the original MNIST/Fashion-MNIST experiments and provides additional evidence on the scalability of the proposed communication-saving mechanisms.
> >
> > **Stronger communication-efficient baselines.**
> >     We now include FedPAQ[1] compressed communication baselines, LAG-FedAMS(\(R=1\)) and LAG-FedAMS(\(R=5\)) as single-step and multi-step lazy aggregation baselines, and LENA as an additional adaptive communication baseline. These baselines allow a more direct comparison with compression-based, lazy-aggregation, and adaptive-communication methods under the same backbone optimizer and partial participation setting.
> >
> > **Ablation of compression and adaptive communication.**
> >     To separate the contribution of compression from that of NLA/AA, we additionally compare FedCAMS, FedNLACA/FedACA, and FedBNLACA/FedBACA. FedCAMS serves as the compression-only baseline with error feedback, FedNLACA/FedACA add one-sided NLA/AA-type adaptive communication, and FedBNLACA/FedBACA further incorporate bidirectional communication reduction.
> >
> > **Direct communication diagnostics.**
> >    We now report direct communication diagnostics in a unified table, including the uplink bit count, downlink bit count, total communication bit count, and the communication cost required to reach the target accuracy. This table also summarizes whether each method uses Top-(k) compression, error feedback, NLA-/AA-/LAG communication mechanisms, and bidirectional communication reduction.
> >
> >
> > **We have incorporated these new experimental protocols and evaluation metrics into the revised Section 5. The specific revisions are highlighted in blue and are accompanied by the corresponding figures and tables.**
> >
> >
> > [1] Reisizadeh, Amirhossein, Aryan Mokhtari, Hamed Hassani, Ali Jadbabaie, and Ramtin Pedarsani. "Fedpaq: A communication-efficient federated learning method with periodic averaging and quantization." In International conference on artificial intelligence and statistics, pp. 2021-2031. PMLR, 2020.

---

> > > ### Author Response · Authors · 2026-06-10
> > >
> > > ## **Minor Concerns 1-3**
> > >
> > > We thank the reviewer for pointing this out. We have carefully checked the manuscript and unified the algorithm names, theorem references, and notation throughout the paper. In particular, we corrected the inconsistent uses of FedLAA, FedLACA, and FedBLACA to FedNLAA, FedNLACA, and FedBNLACA, respectively. We also corrected the theorem references, including the reference to Algorithm 2 in Theorem 6 and the participation setting in Corollary 5. Moreover, we revised the notation in Theorem 5 so that the condition is imposed on the local learning rate $\eta_l$. Finally, we conducted a careful proofreading pass and corrected typographical and grammatical errors throughout the manuscript.
> > >
> > > ## **Minor Concerns 4**
> > >
> > > We appreciate the valuable suggestions from the reviewers. We agree with the original manuscript that
> > > it discussed the selection of strategy parameters C and D, but
> > > it did not provide sufficient empirical evidence to support this guidance. In the revised manuscript, we added a small ablation study on C and D.
> > > Specifically, we evaluated three threshold settings for NLA parameter C
> > > and AA parameter D, corresponding to low threshold, high threshold, and appropriate threshold, respectively.
> > >
> > > **Experimental results are shown in Appendix 1.2.**
> > >
> > > The low threshold setting is more conservative, with limited reduction in traffic,
> > > while the high threshold setting is more aggressive, potentially leading to a poor trade-off between traffic and accuracy,
> > > in contrast, the appropriate threshold
> > > achieves higher test accuracy with fewer uplink bits. These results
> > > empirically validate that C and D should be chosen within a moderate range,
> > > which is consistent with the discussion in the paper.

---

### Review · Reviewer_iAoS · 2026-05-29

**Summary Of Contributions:**

1.Proposes two operationally simple strategies, NLA and AA, which reduce the storage requirement for historical iteration parameters compared to traditional LAG algorithms.

2.Integrates the strategies with AMSGrad, compression, and error feedback to design multiple algorithms covering both one-way and two-way communication.

3.Provides convergence guarantees under non-convex, heterogeneous, and partial client participation settings.

**Audience:**

Yes

**Broader Impact Concerns:**

I did not find any significant ethical concerns that require a Broader Impact Statement.

**Claims And Evidence:**

No

**Requested Changes:**

Critical points:

1、The authors should cite and discuss prior work on dynamic communication thresholds (e.g., AdaComm, Deep Gradient Compression with adaptive threshold, sparsification triggers in gradient compression, stale gradient discarding in asynchronous SGD). They need to clearly articulate the novelty of NLA and AA relative to these existing ideas and provide a direct quantitative comparison with LAG algorithms (Chen et al. 2018; Sun et al. 2019), specifying under what conditions NLA outperforms LAG.

2、The authors should justify whether the linear compressor error relationship in Assumption 4 holds for practical compressors like Top‑k, or revise the assumption accordingly. If it does not hold, the theoretical analysis needs to be re‑examined or restricted to compressors that satisfy the condition.

3、 The empirical evaluation should be extended to larger and more challenging datasets (at least CIFAR‑10/100 or ImageNet) and larger models to demonstrate the scalability of the proposed communication compression methods.

4、Plase include comparisons with more recent or relevant baselines such as FedPAQ, and any other state‑of‑the‑art adaptive compression methods.

5、All experimental results should be reported with error bars (e.g., mean and standard deviation over multiple runs with different random seeds). Single‑run curves are insufficient.

6、Please perform an ablation study to isolate the contributions of the NLA and AA strategies from the compression technique. This is essential to verify the effectiveness of the core proposed strategies.

**Strengths And Weaknesses:**

Pros:

1.NLA and AA rely only on a simple threshold comparison between the parameters of two consecutive steps, making them easy to implement. They can be readily deployed in existing frameworks.

2.The analysis covers full/partial participation, homogeneous/heterogeneous data, and scenarios with/without compression, with convergence rates provided.

Cons:

1.NLA and AA are essentially conditional parameter skipping and conditional momentum accumulation. These ideas have long existed in the areas: "Sparsification trigger conditions" in gradient compression (e.g., transmitting only when gradient change exceeds a threshold),
"Stale gradient discarding" in asynchronous SGD, and "Δ-thresholding" in adaptive communication. The authors do not cite any early work on dynamic communication thresholds, nor do they compare with methods such as AdaComm or Deep Gradient Compression with adaptive threshold.

2.The only essential difference between NLA and the existing LAG algorithm (Chen et al. 2018; Sun et al. 2019) is that the judgment condition changes from "multi-step accumulated error" to "difference between two consecutive steps". The authors claim that "no multiple rounds of parameters are needed", but LAG variants can also use only a single step. A quantitative comparison is missing: under what conditions is NLA better than LAG?

3.Assumption 4 is extremely strong. It requires the compressor error to satisfy a linear relationship before and after aggregation. It is unclear whether this holds for compressors like Top-k.

4.Only two relatively simple image datasets, MNIST and Fashion-MNIST, are used. Evaluations on CIFAR 10/100 or ImageNet are lacking. Moreover, the models used are too small. I believe communication compression should demonstrate scalability on large models.

5.The choice of baseline algorithms is outdated. Comparisons with more advanced or more relevant methods such as FedPAQ are missing.

6.No trade-off curves between compression ratio and final accuracy are shown. Only results for two compression ratios (1/8 and 1/128) are given, and it is not explained why these specific values were chosen.

7.No error bars are shown. all curves come from a single run, making it impossible to assess the impact of randomness.

8.No ablation study is conducted to isolate the contributions of the NLA/AA strategies versus the compression technique, so the effectiveness of the core strategies cannot be verified.

---

> ### Author Response · Authors · 2026-06-10
>
> ##  **Weaknesses 1**
>
> Thank you for the valuable comments from the reviewers.
> We have expanded the "Related Work" section to include AdaComm-type adaptive communication scheduling,
> Deep Gradient Compression (DGC), sparse in-memory SGD, LAG/LAQ-type lazy aggregation, self-triggered upload methods,
> error feedback compression, and handling of obsolete gradients in asynchronous SGD.
>
> We also clarify that NLA and AA are not direct instantiations of these existing rules.
> They are newly defined  communication update strategies that act on
> the objects being transmitted, such as local updates, compressed updates, or
> downlink parameters. NLA eliminates redundant transmissions through
> relative stability testing of the last iteration, while AA further utilizes conditional accumulation to
> utilize stable continuous updates at the communication strategy level. These characteristics distinguish NLA/AA from periodic scheduling, coordinate-level
> sparseness, and obsolete gradient reuse mechanisms. **For details, please refer to the revisions highlighted in blue in the  "Related work" section of the revised manuscript.**

---

> > ### Author Response · Authors · 2026-06-10
> >
> > ## **Weaknesses 2**
> >
> > We sincerely thank the reviewer for this sharp and insightful observation. We
> > agree that our previous explanation was not precise enough. The distinction
> > between NLA and LAG should not be described merely as ``multi-step'' versus
> > ``two consecutive steps'', because, as the reviewer correctly pointed out, LAG
> > can also be specialized to a single-step rule by taking \(R=1\). We have revised the manuscript to clarify this point and ensure precision.  **The specific updates are highlighted in blue within the 'Main contributions' section.**
> >
> > The key difference lies in the quantity controlled by the triggering condition.
> > Even when \(R=1\), LAG remains a stale-gradient discrepancy test. Specifically,
> > the single-step LAG condition can be written as
> > $$
> > \left\|\nabla F_i(\theta_i^{t-1})-\nabla F_i(\theta^t)\right\|^2\leq
> > \frac{\xi_1}{\alpha^2m^2}
> > \left\|
> > \theta^t-\theta^{t-1}
> > \right\|^2 .
> > $$
> > Thus, LAG decides whether the stale gradient information is still sufficiently
> > accurate relative to the recent global model movement. Its smoothness-based
> > version further replaces the gradient discrepancy by the surrogate quantity
> >
> > $$L_i^2\|\theta_i^{t-1}-\theta^t\|^2 .
> > $$
> >
> > By contrast, NLA is a transmitted-object-level relative-stability rule. It checks
> > $$
> > \|x_t^i-x_{t-1}^i\|
> > \leq
> > \frac{C}{\alpha |S_t|}
> > \|x_{t-1}^i\|,
> > $$
> > where $x_t^i$ denotes the object actually communicated in the corresponding
> > algorithm, such as a local model update, a compressed update, or a downlink model
> > parameter. Therefore, NLA directly controls the relative variation of the
> > transmitted object itself, rather than the accuracy of stale gradient
> > information.
> >
> > This distinction provides a concrete condition under which NLA can skip
> > communication while one-step LAG cannot. Specifically, NLA is triggered but the
> > single-step LAG condition is not satisfied when
> > $$
> > \frac{\|x_t^i-x_{t-1}^i\|}{\|x_{t-1}^i\|}\leq
> > \frac{C}{\alpha |S_t|}$$
> > but
> >
> > $$
> > \frac{\|\nabla F_i(\theta_i^{t-1})-\nabla F_i(\theta^t)\|}{\|\theta^t-\theta^{t-1}\|}>\frac{\sqrt{\xi_1}}{\alpha m}.
> > $$
> >
> > This corresponds to the regime where the transmitted object is already relatively
> > stable, whereas the stale-gradient discrepancy used by LAG remains large relative
> > to the recent global movement.
> >
> > Moreover, the replacement error introduced by NLA is directly controlled by its
> > triggering rule. Let $\mathcal M_t^{\mathrm{NLA}}$ denote the set of clients for
> > which NLA is triggered. Then the aggregation perturbation caused by replacing
> > $x_t^i\) with \(x_{t-1}^i$ satisfies
> > $$
> > \left\|
> > \frac{1}{|S_t|}
> > \sum_{i\in\mathcal M_t^{\mathrm{NLA}}}
> > (x_{t-1}^i-x_t^i)
> > \right\|
> > \leq
> > \frac{C}{\alpha |S_t|^2}
> > \sum_{i\in\mathcal M_t^{\mathrm{NLA}}}
> > \|x_{t-1}^i\|.
> > $$
> > Hence, NLA can be more communication-efficient than one-step LAG when the
> > communicated objects are relatively stable, so that the replacement-induced
> > perturbation is small, while the stale-gradient discrepancy condition in LAG is
> > not satisfied. Conversely, when the stale-gradient discrepancy is small but the
> > transmitted object varies substantially, LAG may skip more communication than
> > NLA. Therefore, we do not claim that NLA uniformly dominates LAG.
> >
> > In addition, NLA has an implementation advantage in partial-participation FL. The
> > NLA rule can be evaluated using the last locally stored transmitted object and is
> > naturally compatible with the local error-feedback variables used in our
> > compressed algorithms. Inactive clients can keep their local residuals unchanged,
> > and the rule does not require maintaining a multi-window stale-gradient history.
> > For multi-window LAG implementations, this reduces the amount of historical
> > bookkeeping. Even for \(R=1\), NLA remains different because it monitors the
> > stability of the actually transmitted object, while LAG monitors stale-gradient
> > accuracy.
> >
> >
> > **In the revised manuscript, we have added a concise comparative discussion in Section 3.2 and provided additional quantitative comparisons in Appendix 1.3. Furthermore, we have included empirical comparisons with LAG-FedAMS (\(R=1\)) and multi-step LAG-FedAMS variants under the same backbone optimizer, threshold grid, and partial participation setting.**

---

> > > ### Author Response · Authors · 2026-06-10
> > >
> > > ## **Weaknesses 3**
> > >
> > >
> > > Thanks! We agree that the original
> > > Assumption 4 is stronger than the standard biased-compressor condition. In
> > > particular, deterministic client-wise Top-\(k\) does not automatically satisfy
> > > this assumption in the worst case, because Top-\(k\) is nonlinear and the selected
> > > supports may differ across clients. Consequently, compression and averaging do
> > > not generally commute.
> > >
> > > In the revised manuscript, we have renamed Assumption 4 as a
> > > *compression--aggregation compatibility condition* and clarified its role.
> > > This condition is distinct from the standard contractive biased-compressor
> > > condition
> > > $
> > >     \|C(x)-x\|\le q\|x\|,\qquad q\in[0,1).
> > > $
> > > For example, Top-\(k\) satisfies this standard contractive condition with
> > > \(q=\sqrt{1-k/d}\), but this contraction property alone does not imply the
> > > compression--aggregation compatibility condition. Therefore, our convergence
> > > results for the compressed algorithms should be understood as conditional on this
> > > additional trajectory-level bounded-mismatch assumption. We have revised the
> > > theorem statements accordingly to clarify that the theoretical guarantees hold
> > > for compressors and training trajectories satisfying this compatibility
> > > condition.
> > >
> > > To make this issue explicit, we added a discussion explaining why Top-\(k\) does
> > > not satisfy the compatibility condition automatically. The main reason is that
> > > the support selected by Top-\(k\) before aggregation can differ from the support
> > > selected after aggregation. Hence, in general,
> > > $$
> > >     C\left(\frac1{|S_t|}\sum_{i\in S_t}z_t^i\right)
> > >     \neq
> > >     \frac1{|S_t|}\sum_{i\in S_t}C(z_t^i),
> > >     \qquad z_t^i=\Delta_t^i+e_t^i .
> > > $$
> > > Thus, Assumption 4 should not be interpreted as a generic algebraic property of
> > > Top-\(k\), but as a compatibility condition controlling the mismatch along the
> > > optimization trajectory.
> > >
> > > To examine whether this condition is reasonable for the compressor used in our
> > > experiments, we added an empirical diagnostic. Specifically, for the participating
> > > client set \(S_t\), we report the trajectory-level mismatch ratio
> > >
> > > $$
> > > \gamma_t=\frac{\left\|C\left(|S_t|^{-1}\sum_{i\in S_t} z_t^i\right)-|S_t|^{-1}\sum_{i\in S_t} C(z_t^i)\right\|}{\left\||S_t|^{-1}\sum_{i\in S_t}\Delta_t^i\right\|+\varepsilon},
> > > z_t^i=\Delta_t^i+e_t^i ,
> > > $$
> > >
> > > where $\varepsilon>0$ is a small numerical constant used to avoid division by
> > > zero.
> > >
> > > We also report the second compatibility ratio associated with the NLA/AA
> > > triggering term. When $M_t\neq\varnothing$, where $M_t$ denotes the set of
> > > clients satisfying the NLA/AA judgment condition, we compute
> > >
> > > $$
> > > \lambda_t=\frac{\left\||M_t|^{-1}\sum_{i\in M_t} C(q_t^i)\right\|}{\left\||S_t|^{-1}\sum_{i\in S_t}\Delta_t^i\right\|+\varepsilon}.$$
> > >
> > > When $M_t=\varnothing$, no triggering perturbation is introduced in that round,
> > > and the corresponding ratio is omitted.
> > >
> > > The empirical results show that both \(\gamma_t\) and \(\lambda_t\) remain bounded
> > > along the observed training trajectories for the Top-\(k\) compressor used in our
> > > experiments. In particular, $\lambda_t$ remains relatively small, while
> > > $\gamma_t$ may exhibit occasional spikes but stays finite along the training
> > > path. These empirical results do not claim that Assumption 4 holds for arbitrary
> > > Top-\(k\) in the worst case. Rather, they support the interpretation of
> > > Assumption 4 as a trajectory-level bounded-mismatch condition in our experimental
> > > setting.
> > >
> > > Accordingly, we have revised the manuscript to avoid stating or implying that
> > > Assumption 4 is automatically satisfied by Top-\(k\). **\bf We added the clarification
> > > after Definition 1 and included the empirical validation of  $\gamma_t$ and
> > > $\lambda_t$ in the Appendix1.1.**

---

> > > > ### Author Response · Authors · 2026-06-10
> > > >
> > > > ## **Weaknesses 4-8**
> > > >
> > > >  We thank the reviewer for these constructive comments. We agree that the
> > > > original experimental section did not sufficiently demonstrate scalability,
> > > > baseline strength, compression--accuracy trade-offs, robustness to randomness,
> > > > and the individual contributions of NLA/AA. In the revised manuscript, we have
> > > > therefore substantially expanded the experimental evaluation. The main changes
> > > > are summarized below.
> > > >
> > > > **Larger dataset and model.**
> > > >     To address the concern that only MNIST and Fashion-MNIST were used, we have
> > > >     added new experiments on CIFAR-10 with a VGG-11 model under the non-IID
> > > >     federated setting. This setting is substantially more challenging than the
> > > >     original MNIST/Fashion-MNIST experiments and is used to evaluate whether the
> > > >     proposed communication-saving mechanisms scale to deeper convolutional
> > > >     models and more complex image data.
> > > >
> > > > **Stronger and more relevant baselines.**
> > > >     We have added FedPAQ[1] compressed communication baselines and LAG-type
> > > >     adaptive communication baselines. In particular, we include
> > > >     LAG-FedAMS(\(R=1\)) and LAG-FedAMS(\(R=5\)) to directly compare with
> > > >     single-step and multi-step lazy aggregation mechanisms. We also include
> > > >     LENA as an additional adaptive communication baseline. These additions make
> > > >     the comparison more directly aligned with communication-efficient federated
> > > >     optimization.
> > > >
> > > > **Compression-ratio and accuracy trade-off.**
> > > >     We now provide explicit communication--accuracy trade-off curves under
> > > >     different compression levels, including Top-\(k\) ratios \(1/8\) and
> > > >     \(1/128\). We clarify that \(1/8\) represents a moderate compression regime,
> > > >     while \(1/128\) represents an aggressive compression regime. These two
> > > >     settings are used to evaluate whether the proposed NLA/AA mechanisms remain
> > > >     effective under both mild and severe communication constraints.
> > > >
> > > > **Error bars and randomness.**
> > > >     All main curves in the revised experiments are now averaged over three
> > > >     independent random seeds. The solid curves denote the mean trajectory and the
> > > >     shaded regions denote one standard deviation. Since adaptive communication
> > > >     may lead to different communication budgets across seeds, we interpolate all
> > > >     trajectories onto a common communication-bit grid before averaging.
> > > >
> > > > **Ablation study.**
> > > >     We have added a component-wise ablation study to separate the effects of
> > > >     Top-\(k\) compression, error feedback, NLA/AA, and bidirectional
> > > >     communication reduction. The new ablation table compares FedAvg, FedAdam,
> > > >     FedAMS, LENA, LAG-FedAMS, FedPAQ, FedCAMS, FedNLACA/FedACA, and
> > > >     FedBNLACA/FedBACA variants. This allows us to verify that the observed
> > > >     communication savings are not solely due to compression, but also come from
> > > >     the proposed adaptive upload/download mechanisms.
> > > >
> > > >
> > > > In addition, we now report direct communication diagnostics, including uplink
> > > > bits, downlink bits, total bits,
> > > > and the number of communication bits required to reach a target accuracy.  **These
> > > > new results are reported in the revised 5 experimental section.**
> > > >
> > > >
> > > > [1] Reisizadeh, Amirhossein, Aryan Mokhtari, Hamed Hassani, Ali Jadbabaie, and Ramtin Pedarsani. "Fedpaq: A communication-efficient federated learning method with periodic averaging and quantization." In International conference on artificial intelligence and statistics, pp. 2021-2031. PMLR, 2020.

---

> > > > > ### Author Response · Authors · 2026-06-10
> > > > >
> > > > > ## **Requested Changes 1**
> > > > > See Weaknesses 1 and 2 for details.
> > > > >
> > > > > ## **Requested Changes 2**
> > > > > See Weaknesses 3 for details.
> > > > >
> > > > > ## **Requested Changes 3-6**
> > > > > See Weaknesses 4-8 for details.

---

### Decision · Action_Editor_opWk · 2026-07-13

**Recommendation:** Accept as is

**Audience:**

Yes

**Claims And Evidence:**

I make acceptance.